# Antibiotic potentiation and inhibition of cross-resistance in pathogens associated with cystic fibrosis

**Nikol Kadeřábková**[1,2]*[†], **R Christopher D Furniss**[2][†], **Evgenia Maslova**[3],
**Kathryn E Potter**[1], **Lara Eisaiankhongi**[3], **Patricia Bernal**[4], **Alain Filloux**[2,5,6,7],
**Cristina Landeta**[8], **Diego Gonzalez**[9], **Ronan R McCarthy**[3],
**Despoina AI Mavridou**[1,10]*

[1]Department of Molecular Biosciences, The University of Texas at Austin, Austin, United States; [2]Centre for Bacterial Resistance Biology, Department of Life Sciences, Imperial College London, London, United Kingdom; [3]Division of Biosciences, Department of Life Sciences, College of Health and Life Sciences, Brunel University London, Uxbridge, United Kingdom; [4]Departamento de Microbiología, Facultad de Biología, Universidad de Sevilla, Seville, Spain; [5]Singapore Centre for Environmental Life Sciences Engineering, Nanyang Technological University, Singapore, Singapore; [6]School of Biological Sciences, Nanyang Technological University, Singapore, Singapore; [7]Lee Kon Chian School of Medicine, Nanyang Technological University, Singapore, Singapore; [8]Department of Biology, Indiana University, Bloomington, United States; [9]Laboratoire de Microbiologie, Institut de Biologie, Université de Neuchâtel, Neuchâtel, Switzerland; [10]John Ring LaMontagne Center for Infectious Diseases, The University of Texas at Austin, Austin, United States

**\*For correspondence:**
nikol.kaderabkova@austin.utexas.edu (NK);
despoina.mavridou@austin.utexas.edu (DAIM)

[†]These authors contributed equally to this work

**Competing interest:** The authors declare that no competing interests exist.

## eLife Assessment

This **important** study demonstrates that disruption of a common protein-folding system renders drug-resistant clinical bacteria susceptible to antibiotics. The work **convincingly** shows that targeting protein folding, including through therapeutic use of small-molecule inhibitors, can be used to combat multidrug-resistant pathogens by potentiating the efficacy of existing drugs. This study is significant and timely as it informs on a new strategy that is relevant to microbiologists and clinicians interested in combating antimicrobial resistance.

**Abstract** Critical Gram-negative pathogens, like *Pseudomonas*, *Stenotrophomonas*, and *Burkholderia*, are now resistant to most antibiotics. Complex resistance profiles, together with synergistic interactions between these organisms, increase the likelihood of treatment failure in distinct infection settings, for example in the lungs of cystic fibrosis (CF) patients. Here, we discover that cell envelope protein homeostasis pathways underpin both antibiotic resistance and cross-protection in CF-associated bacteria. We find that inhibition of oxidative protein folding inactivates multiple species-specific resistance proteins. Using this strategy, we sensitize multidrug-resistant *Pseudomonas aeruginosa* to β-lactam antibiotics and demonstrate promise of new treatment avenues for the recalcitrant emerging pathogen *Stenotrophomonas maltophilia*. The same approach also inhibits cross-protection between resistant *S. maltophilia* and susceptible *P. aeruginosa*, allowing eradication of both commonly co-occurring CF-associated organisms. Our results provide the basis for the development of next-generation strategies that target antibiotic resistance, while also impairing specific interbacterial interactions that enhance the severity of polymicrobial infections.

## Introduction

Antimicrobial resistance (AMR) is one of the most significant threats to health systems worldwide (*Rochford et al., 2018*). Since the end of the 'golden age' of antibiotic discovery in the 1970s, very few new antimicrobial agents have entered the clinic, and most of those that have gained approval are derivatives of existing antibiotic classes (*Baker et al., 2018*; *Brown and Wright, 2016*; *Chopra, 2013*). Meanwhile, resistance to useful antibiotics is continuously rising, resulting in more than 1.3 million deaths annually (*Antimicrobial Resistance, 2022*). In addition to the undeniable surge of resistance, it is becoming apparent that intra- and inter-species interactions also play a role in AMR and its evolution (*Bottery et al., 2021*), ultimately posing additional challenges during antibiotic treatment. This necessitates not only the development of novel antimicrobials and strategies that will expand the lifespan of existing antibiotics but also the implementation of approaches that will address the polymicrobial nature of most infections.

Antibiotic resistance is most commonly evaluated by testing bacterial strains in monoculture. Nonetheless, the majority of clinical infections contain multiple species whose coexistence in complex pathobionts often limits our treatment options. This is of particular importance for recalcitrant infections such as the polymicrobial communities found in the lungs of cystic fibrosis (CF) patients. CF lung infections have become a paradigm for chronic infectious diseases that result in poor quality of life and early patient mortality (*Shteinberg et al., 2021*). Such infections are dominated by highly resistant opportunistic pathogens, including, but not limited to, *Pseudomonas aeruginosa*, *Staphylococcus aureus*, species and strains belonging to the *Burkholderia* complex, and *Stenotrophomonas maltophilia* (*Widder et al., 2022*). Most of these organisms carry an array of resistance mechanisms, like efflux pumps, atypical lipopolysaccharide structures, and β-lactamase enzymes. Their co-occurrence in the CF lung leads to treatment challenges since common clinical care options for one pathogen are not necessarily compatible with the antibiotic susceptibility profiles of other species that are present. For example, on the one hand, *P. aeruginosa* is the most prevalent organism in CF lung infections and its treatment, especially during pulmonary exacerbation episodes, relies heavily on β-lactam compounds (*Widder et al., 2022*). On the other hand, CF microbiomes are increasingly found to encompass *S. maltophilia* (*Widder et al., 2022*; *de Vrankrijker et al., 2010*; *Terlizzi et al., 2023*), a globally distributed opportunistic pathogen that causes serious nosocomial respiratory and bloodstream infections (*Alsuhaibani et al., 2021*; *Amin et al., 2020*; *Brooke, 2012*). *S. maltophilia* is one of the most prevalent emerging pathogens (*Amin et al., 2020*), and it is intrinsically resistant to almost all antibiotics, including β-lactams like penicillins, cephalosporins, and carbapenems, as well as macrolides, fluoroquinolones, aminoglycosides, chloramphenicol, tetracyclines, and colistin. As a result, the standard treatment option for lung infections, that is broad-spectrum β-lactam antibiotic therapy, is rarely successful in countering *S. maltophilia* (*Brooke, 2012*; *Calvopiña et al., 2017*), creating a definitive need for approaches that will be effective in eliminating both pathogens.

The lack of suitably broad antibiotic regimes able to simultaneously eradicate all pathogens present in specific infection settings is not the only challenge when treating polymicrobial communities. Bacterial interactions between antibiotic-resistant and antibiotic-susceptible bacteria can add to this problem by adversely affecting antibiotic drug sensitivity profiles of organisms that should be treatable (*Bottery et al., 2021*). In particular, some antibiotic resistance proteins, like β-lactamases, which decrease the quantities of active drug present, function akin to common goods, since their benefits are not limited to the pathogen that produces them but can be shared with the rest of the bacterial community. This means that their activity enables pathogen cross-resistance when multiple species are present (*Bottery et al., 2022*; *Semenec et al., 2023*), something that was demonstrated in recent work investigating the interactions between pathogens that naturally co-exist in CF infections. More specifically, it was shown that in laboratory co-culture conditions, highly drug-resistant *S. maltophilia* strains actively protect susceptible *P. aeruginosa* from β-lactam antibiotics (*Bottery et al., 2022*). Moreover, this cross-protection was found to facilitate, at least under specific conditions, the evolution of β-lactam resistance in *P. aeruginosa* (*Quinn et al., 2022*). The basis of such interactions could be exploited during the design of novel therapeutic strategies, since targeting appropriate resistance enzymes will not only render their producers susceptible to existing drugs but should also impair their capacity to protect co-existing antibiotic-susceptible strains.

Protein homeostasis in the Gram-negative cell envelope, and in particular the formation of disulfide bonds by the thiol oxidase DsbA (*Bardwell et al., 1991*; *Denoncin and Collet, 2013*; *Hiniker and*

*Bardwell, 2004*; *Kadokura et al., 2004*; *Martin et al., 1993*), is essential for the function of many resistance proteins (*Furniss et al., 2022*). Oxidative protein folding occurs post-translationally, after translocation of the nascent polypeptide to the periplasm through the general secretion (Sec) system (*Heras et al., 2007*). There, disulfide bond formation assists the assembly of 40% of the cell-envelope proteome (*Dutton et al., 2008*; *Vertommen et al., 2008*), promotes the biogenesis of virulence factors (*Heras et al., 2009*; *Landeta et al., 2018*), controls the awakening of bacterial persister cells (*Wilmaerts et al., 2019*), and underpins the function of resistance determinants, including enzymes for which we do not currently have inhibitor compounds, such as metallo-β-lactamases (*Tooke et al., 2019*). Here, we reveal the potential of targeting proteostasis pathways, such as disulfide bond formation, as a strategy against pathogens commonly associated with highly resistant polymicrobial infections. Using this approach, we incapacitate species-specific resistance proteins in CF-associated bacteria and simultaneously abrogate protective effects between pathogens that coexist in these infections. Our results demonstrate that such strategies generate compatible treatment options for recalcitrant CF pathogens and, at the same time, eradicate interspecies interactions that impose additional challenges during antibiotic treatment in complex infection settings.

## Results

### Species-specific cysteine-containing β-lactamases depend on oxidative protein folding

**β-lactamase activity.** To investigate the potential of targeting disulfide bond formation as a strategy to overcome resistance mechanisms in challenging pathogens, we chose to primarily explore β-lactamases that are produced by bacteria intimately associated with CF lung infections. DsbA dependence has been previously shown for a handful of such enzymes (*Furniss et al., 2022*), like the chromosomally encoded class B3 metallo-β-lactamase L1-1 from *S. maltophilia* (*Supplementary file 6, supplementary table 1*), which contributes significantly to AMR in this organism (*Brooke, 2012*), as well as β-lactamases from the GES and OXA families, which are broadly disseminated, but commonly found in *P. aeruginosa* (*Hammoudi Halat and Ayoub Moubareck, 2022*; *Yoon and Jeong, 2021*). Here, we selected six clinically important β-lactamases from different Ambler classes (classes A, B, and D) that are exclusively encoded either by *P. aeruginosa* or by the *Burkholderia* complex. The *P. aeruginosa* enzymes (BEL-1, CARB-2, AIM-1, and OXA-50) are all phylogenetically distinct, while the *Burkholderia* β-lactamases (BPS-1m and BPS-6) belong to the same phylogenetic class (*Supplementary file 1*). Class A, C, and D β-lactamases, like the BPS-6 (class A) and OXA-50 (class D) enzymes investigated here, are serine-dependent hydrolases. Serine β-lactamases are structurally related to penicillin binding proteins, which have a major role in the synthesis of the peptidoglycan (*Ambler, 1980*). By contrast, class B enzymes are evolutionarily distinct and rely on one or two $Zn^{2+}$ ions for catalytic activity (*Tooke et al., 2019*; *Hong et al., 2015*). In addition to belonging to different phylogenetic classes, the selected enzymes have different numbers of cysteines, display varied hydrolytic activities, can be both resident on the chromosome or on mobile genetic elements, and have diverse inhibitor susceptibility profiles (*Supplementary file 6, supplementary table 1*).

We expressed all six β-lactamases in the *Escherichia coli* K-12 strain MC1000 and its isogenic *dsbA* deletion mutant. This strain background was selected because it has been traditionally used in oxidative protein folding studies (*Landeta et al., 2015*; *Mössner et al., 1999*; *Rietsch et al., 1997*; *Stewart et al., 1999*) and it lacks endogenous β-lactamase enzymes or any other mechanisms that could contribute to antibiotic resistance. We recorded β-lactam minimum inhibitory concentration (MIC) values for each enzyme in both strain backgrounds. We found that expression of all test enzymes in the *dsbA* mutant background resulted in markedly reduced MICs for at least one β-lactam antibiotic (*Figure 1* and *Supplementary file 2A*), compared to the MICs recorded in the wild-type *E. coli* strain; only differences larger than twofold were considered. These results indicate that the presence of DsbA is important for the function of all tested resistance proteins.

To ensure that effects shown in *Figure 1* are not due to factors that are not specific to the interaction of DsbA with the tested β-lactamases, we also performed a series of control experiments. We have previously shown that deletion of *dsbA* does not affect the aerobic growth of *E. coli* MC1000, or the permeability of its outer and inner membranes (*Furniss et al., 2022*). Furthermore, here we observed no changes in MIC values for the aminoglycoside antibiotic gentamicin, which is not

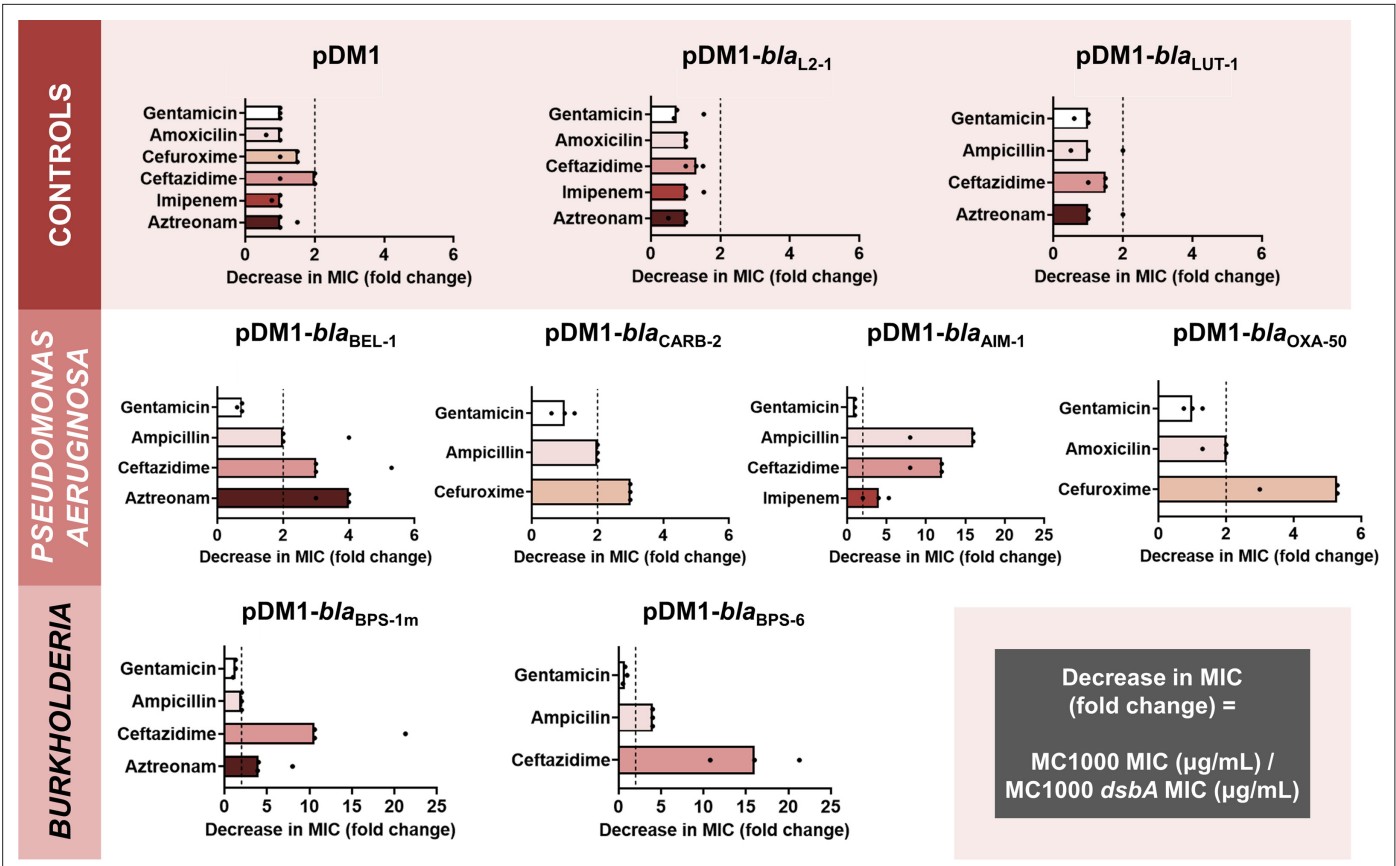

**Figure 1.** The function of species-specific cysteine-containing β-lactamases from cystic fibrosis-associated pathogens depends on DsbA-mediated oxidative protein folding. β-lactam MIC values for *E. coli* MC1000 expressing diverse disulfide-bond-containing β-lactamases (Ambler classes A, B, and D) are substantially reduced in the absence of DsbA (MIC fold changes: >2; fold change of 2 is indicated by the black dotted lines). No changes in MIC values are observed for the aminoglycoside antibiotic gentamicin (white bars), confirming that absence of DsbA does not compromise the general ability of this strain to resist antibiotic stress. Minor changes in MIC values (≤2 fold) are observed for strains harboring the empty vector control (pDM1) or those expressing the class A β-lactamases L2-1 and LUT-1, which contain two or more cysteines (*Supplementary file 6, supplementary table 1*), but no disulfide bonds (top row). Graphs show MIC fold changes for β-lactamase-expressing *E. coli* MC1000 and its *dsbA* mutant from three biological experiments each conducted as a single technical repeat; the MIC values used to generate this figure are presented in *Supplementary file 2A* (rows 2–7 and 9–20).

The online version of this article includes the following figure supplement(s) for figure 1:

**Figure supplement 1.** Complementation of *dsbA* restores the β-lactam MIC values for *E. coli* MC1000 *dsbA* expressing β-lactamase enzymes.

degraded by β-lactamases, or between the parental *E. coli* strain and its *dsbA* mutant harboring only the empty vector (*Figure 1* and *Supplementary file 2A*). In addition, *E. coli* strains expressing either of two disulfide-free enzymes, the class A β-lactamases L2-1 and LUT-1 from *S. maltophilia* and *Pseudomonas luteola*, respectively, did not exhibit decreased MICs in the absence of *dsbA* (*Figure 1* and *Supplementary file 2A*). These proteins were selected because they both contain two or more cysteine residues but lack disulfide bonds due to the fact that they are transported to the periplasm, pre-folded, by the Twin-arginine translocation (Tat) pathway, rather than by the Sec system. In the case of L2-1, Tat-dependent transport has been experimentally confirmed (*Pradel et al., 2009*), whilst LUT-1 contains a predicted Tat signal sequence (SignalP 5.0 (*Almagro Armenteros et al., 2019*) likelihood scores: Sec/SPI = 0.0572, Tat/SPI = 0.9312, Sec/SPII (lipoprotein)=0.0087, other = 0.0029). Finally, the specific interaction between DsbA and our selected test enzymes was further supported by the fact that complementation of *dsbA* generally restores MICs to near wild-type values for the latest generation β-lactam that each β-lactamase can hydrolyze (*Figure 1—figure supplement 1*); we only achieve partial complementation for the *dsbA* mutant expressing BPS-1m, which we attribute to the fact that expression of this enzyme in *E. coli* is sub-optimal.

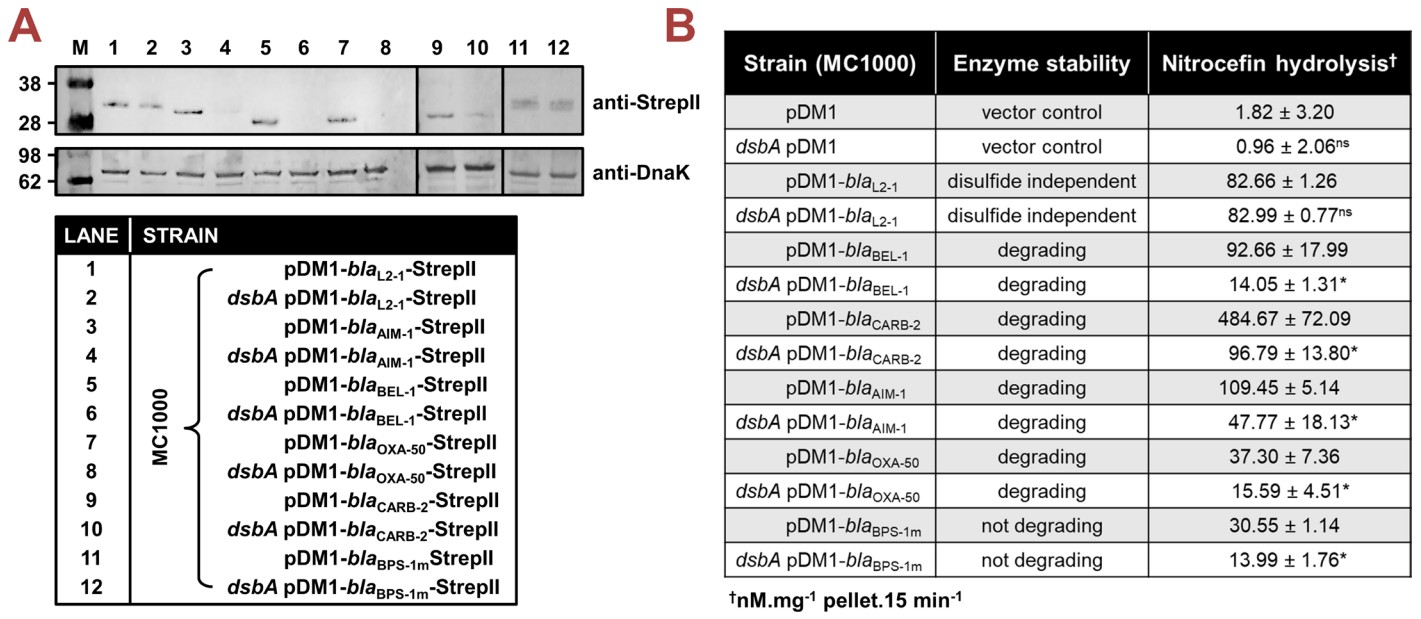

**Figure 2.** Absence of DsbA results in degradation or misfolding of species-specific cysteine-containing β-lactamases. (**A**) The protein levels of most tested disulfide-bond-containing Ambler class A, B, and D β-lactamases are drastically reduced when these enzymes are expressed in *E. coli* MC1000 *dsbA*; the amount of the control enzyme L2-1, containing three cysteines but no disulfide bonds, is unaffected. An exception to this is the class A enzyme BPS-1m, for which no decrease in abundance is observed in the *dsbA* mutant (compare lanes 11 and 12). Protein levels of StrepII-tagged β-lactamases were assessed using a Strep-Tactin-AP conjugate. A representative blot from three biological experiments, each conducted as a single technical repeat, is shown; molecular weight markers (M) are on the left, DnaK was used as a loading control and solid black lines indicate where the membrane was cut. Full immunoblots and SDS-PAGE analysis of the immunoblot samples for total protein content are shown in *Supplementary file 3*. (**B**) The hydrolysis of the chromogenic β-lactam nitrocefin by cysteine-containing β-lactamases is impaired when these enzymes are expressed in *E. coli* MC1000 *dsbA*. The hydrolytic activities of strains harboring the empty vector or expressing the control enzyme L2-1 show no dependence on DsbA. The 'Enzyme stability' column informs on the abundance of each enzyme when it is lacking its disulfide bond(s); this was informed from the immunoblotting experiments in panel (A) The 'Nitrocefin hydrolysis' column shows the amount of nitrocefin hydrolyzed per mg of bacterial cell pellet in 15 min. n=3, table shows means ± SD, significance is indicated by *=p < 0.05, ns = non-significant.

The online version of this article includes the following source data for figure 2:

**Source data 1.** Original files of the full raw unedited immunoblots used to prepare *Figure 2A*.

**Source data 2.** Uncropped immunoblots used to prepare *Figure 2A*.

Taken together, our data show that DsbA-mediated disulfide bond formation is important for the function of all tested, species-specific β-lactamases. Of these, the most affected enzymes (largest MIC value decreases; *Figure 1* and *Supplementary file 2A*) are the class A extended-spectrum-β-lactamases (ESBLs) from *Burkholderia* (BPS-1m and BPS-6) and the class B3 metallo-β-lactamase AIM-1, which, like all other class B enzymes (*Ju et al., 2018*), is resistant to inhibition by classical β-lactamase inhibitor compounds (*Supplementary file 6, supplementary table 1*; *Tooke et al., 2019*).

**β-Lactamase abundance and folding.** To gain insight into how impairment of disulfide bond formation impacts the production or activity of the tested enzymes (*Figure 1*), we first performed immunoblotting for all phylogenetically distinct β-lactamases (AIM-1, BEL-1, OXA-50, CARB-2, and BPS-1m) to assess their protein levels in the presence and absence of *dsbA*. For four of the five tested β-lactamases (AIM-1, BEL-1, OXA-50, and CARB-2), deletion of *dsbA* resulted in drastically reduced protein levels compared to the levels of the control enzyme L2-1, which remained largely unaffected (*Figure 2A*). This shows that without their disulfide bonds, these proteins are unstable and are ultimately degraded by other cell envelope proteostasis components (*Goemans et al., 2014*). This was further corroborated by the fact that lysates from *dsbA* mutants expressing these four enzymes showed significantly reduced hydrolytic activity towards the chromogenic β-lactamase substrate nitrocefin (*Figure 2B*). In the case of BPS-1m, enzyme levels were unchanged in the absence of *dsbA* (*Figure 2A*). However, without its disulfide bond, this protein was significantly less able to hydrolyze nitrocefin (*Figure 2B*), suggesting a folding defect that results in loss of function. The latter is consistent

with the reduced MICs conferred by BPS-1m (and its sister enzyme BPS-6) in the absence of *dsbA* (*Figure 1*). The data presented so far (*Figures 1 and 2*) demonstrate that disulfide bond formation is essential for the biogenesis (stability and/or protein folding) and, in turn, activity of an expanded set of clinically important β-lactamases, including enzymes that currently lack inhibitor options.

## Targeting oxidative protein folding inhibits both antibiotic resistance and interbacterial interactions in CF-associated pathogens

**Sensitization of multidrug-resistant *P. aeruginosa* clinical isolates**. The efficacy of commonly used treatment options against *P. aeruginosa* in CF lung infections, namely piperacillin-tazobactam and cephalosporin-avibactam combinations, as well as more advanced drugs like aztreonam or carbapenems (*George et al., 2009*; *Kunz Coyne et al., 2022*), is increasingly threatened by an array of β-lactamases, encompassing both broadly disseminated enzymes and species-specific ones (*George et al., 2009*; *Kunz Coyne et al., 2022*; *Haines et al., 2022*). To determine whether the effects on β-lactam MICs observed in our inducible system (*Figure 1* and *Furniss et al., 2022*) can be reproduced in the presence of other resistance determinants in a natural context with endogenous enzyme expression levels, we deleted the principal *dsbA* gene, *dsbA1*, in several multidrug-resistant (MDR) *P. aeruginosa* clinical strains (*Supplementary file 6, supplementary table 2*). Pathogenic bacteria often encode multiple DsbA analogues (*Heras et al., 2009*; *Landeta et al., 2018*) and *P. aeruginosa* is no exception. It encodes two DsbAs, but DsbA1 has been found to catalyze the vast majority of the oxidative protein folding reactions taking place in its cell envelope (*Arts et al., 2013*).

We first tested two clinical isolates (strains G4R7 and G6R7; *Supplementary file 6, supplementary table 2*) expressing the class B3 metallo-β-lactamase AIM-1, for which we recorded reduced activity in an *E. coli dsbA* background (*Figures 1 and 2*). This enzyme confers high-level resistance to piperacillin-tazobactam and the third-generation cephalosporin ceftazidime, both anti-pseudomonal β-lactams that are used in the treatment of critically ill patients (*Yong et al., 2012*). Notably, while specific to the *P. aeruginosa* genome, *aim-1* is flanked by two ISCR15 elements suggesting that it remains mobilizable (*Yong et al., 2012*; *Supplementary file 6, supplementary table 1*). MICs for piperacillin-tazobactam and ceftazidime were determined for both AIM-1-positive *P. aeruginosa* isolates and their *dsbA1* mutants (*Figure 3AB*). Deletion of *dsbA1* from *P. aeruginosa* G4R7 resulted in a substantial decrease in its piperacillin-tazobactam MIC value by 192 μg/mL and sensitization to ceftazidime (*Figure 3A*), while the *dsbA1* mutant of *P. aeruginosa* G6R7 became susceptible to both antibiotic treatments (*Figure 3B*). Despite the fact that *P. aeruginosa* G4R7 *dsbA1* was not sensitized for piperacillin-tazobactam, possibly due to the high level of piperacillin-tazobactam resistance of the parent clinical strain, our results across these two isolates show promise for DsbA as a target against β-lactam resistance in *P. aeruginosa*. To further test our approach in an infection context, we performed *in vivo* survival assays using the wax moth model *Galleria mellonella* (*Figure 3C*), an informative non-vertebrate system for the study of new antimicrobial approaches against *P. aeruginosa* (*Hill et al., 2014*). Larvae were infected with *P. aeruginosa* G6R7 or its *dsbA1* mutant, and infections were treated once with piperacillin at a final concentration below the EUCAST breakpoint, as appropriate. No larvae survived beyond 20 hr post-infection when infected with *P. aeruginosa* G6R7 or its *dsbA1* mutant without antibiotic treatment (*Figure 3C*; blue and light blue survival curves). Despite this clinical strain being resistant to piperacillin *in vitro* (*Figure 3B*), treatment with piperacillin *in vivo* increased larval survival (52.5% survival at 28 hr post-infection) compared to the untreated conditions (*Figure 3C*; blue and light blue survival curves) possibly due to *in vivo* ceftazidime MIC values being discrepant to the value recorded *in vitro*. Nonetheless, treatment of *P. aeruginosa* G6R7 *dsbA1* with piperacillin resulted in a significant improvement in survival (77.5% survival at 28 hr post-infection), highlighting increased relative susceptibility compared to the treated wild-type condition (*Figure 3C*; compare the red and pink survival curves).

Next, we tested two *P. aeruginosa* clinical isolates (strains CDC #769 and CDC #773; *Supplementary file 6, supplementary table 2*), each expressing two class A enzymes from the GES family (GES-19/GES-26 or GES-19/GES-20), for which we have previously demonstrated DsbA dependence (*Furniss et al., 2022*). The GES family comprises 59 distinct ESBLs (*Supplementary file 1*), which are globally disseminated and commonly found in *P. aeruginosa*, as well as other critical Gram-negative pathogens (for example *Klebsiella pneumoniae* and *Enterobacter cloacae*) (*Weldhagen, 2006*). Deletion of *dsbA1* in these clinical strains resulted in sensitization to piperacillin-tazobactam and

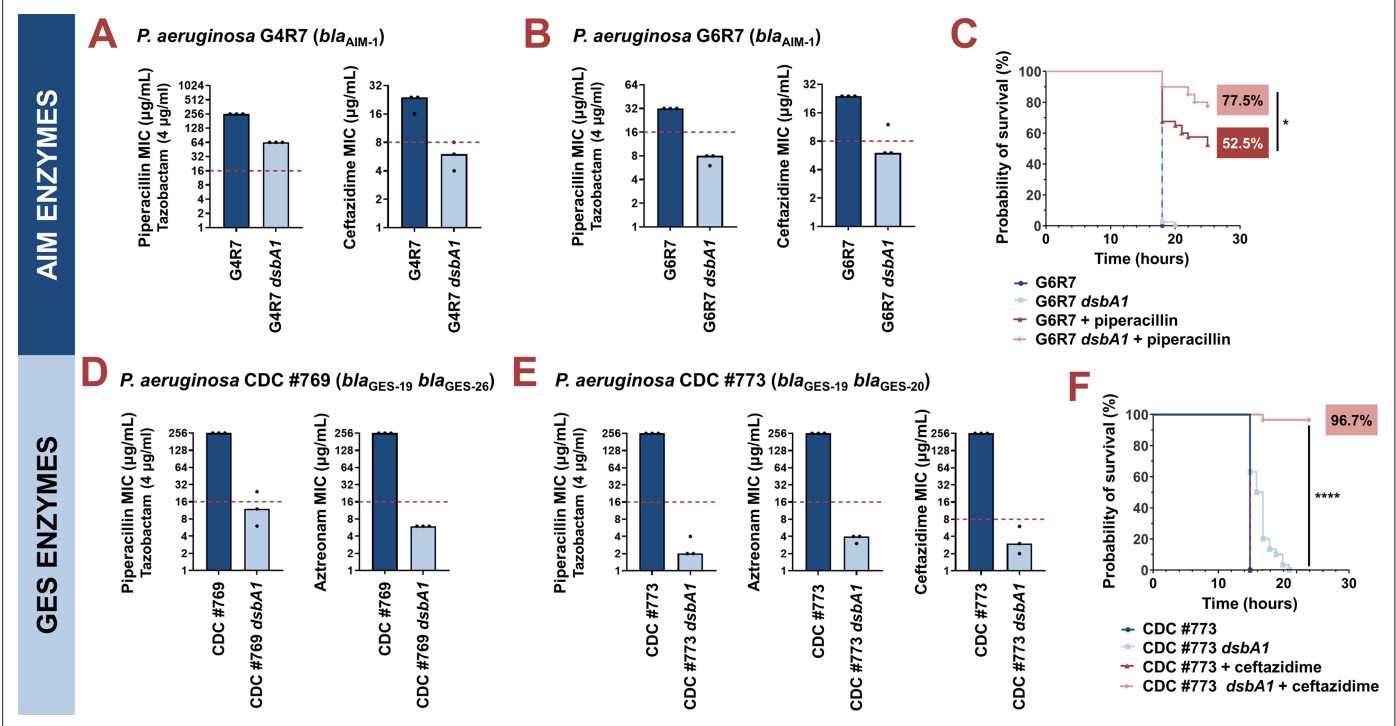

**Figure 3.** Absence of the principal DsbA analogue (DsbA1) allows treatment of multidrug-resistant *Pseudomonas aeruginosa* clinical isolates with existing β-lactam antibiotics. (**A**) Deletion of *dsbA1* in the AIM-1-expressing *P. aeruginosa* G4R7 clinical isolate sensitizes this strain to ceftazidime and results in reduction of the piperacillin/tazobactam MIC value by 192 µg/mL. (**B**) Deletion of *dsbA1* in the AIM-1-expressing *P. aeruginosa* G6R7 clinical isolate sensitizes this strain to piperacillin/tazobactam and ceftazidime. (**C**) 100% of the *G. mellonella* larvae infected with *P. aeruginosa* G6R7 (blue curve) or *P. aeruginosa* G6R7 *dsbA1* (light blue curve) die 18 hr post-infection, while only 52.5% of larvae infected with *P. aeruginosa* G6R7 and treated with piperacillin (red curve) survive 28 hr post-infection. Treatment of larvae infected with *P. aeruginosa* G6R7 *dsbA1* with piperacillin (pink curve) results in 77.5% survival, 28 hr post-infection. The graph shows Kaplan-Meier survival curves of infected *G. mellonella* larvae after different treatment applications; horizontal lines represent the percentage of larvae surviving at the indicated time point (a total of 40 larvae were used for each curve). Statistical analysis of this data was performed using a Mantel-Cox test. The most relevant comparison is noted on the figure. Full statistical analysis is as follows: n=40; p=0.3173 (non-significance; *P. aeruginosa* vs *P. aeruginosa dsbA1*), p<0.0001 (significance; *P. aeruginosa* vs *P. aeruginosa* treated with piperacillin), p<0.0001 (significance; *P. aeruginosa dsbA1* vs *P. aeruginosa* treated with piperacillin), p=0.0147 (significance; *P. aeruginosa* treated with piperacillin vs *P. aeruginosa dsbA1* treated with piperacillin). (**D**) Deletion of *dsbA1* in the GES-19/GES-26-expressing *P. aeruginosa* CDC #769 clinical isolate sensitizes this strain to piperacillin/tazobactam and aztreonam. (**E**) Deletion of *dsbA1* in the GES-19/GES-20-expressing *P. aeruginosa* CDC #773 clinical isolate sensitizes this strain to piperacillin/tazobactam, aztreonam, and ceftazidime. (**F**) 100% of *G. mellonella* larvae infected with *P. aeruginosa* CDC #773 (blue curve), *P. aeruginosa* CDC #773 *dsbA1* (light blue curve) or larvae infected with *P. aeruginosa* CDC #773 and treated with ceftazidime (red curve) die 21 hr post-infection. Treatment of larvae infected with *P. aeruginosa* CDC #773 *dsbA1* with ceftazidime (pink curve) results in 96.7% survival, 24 hr post-infection. The graph shows Kaplan-Meier survival curves of infected *G. mellonella* larvae after different treatment applications; horizontal lines represent the percentage of larvae surviving at the indicated time point (a total of 30 larvae were used for each curve). Statistical analysis of this data was performed using a Mantel-Cox test. The most relevant comparison is noted on the figure. Full statistical analysis is as follows: n=30; p<0.0001 (significance; *P. aeruginosa* vs *P. aeruginosa dsbA1*), p>0.9999 (non-significance; *P. aeruginosa* vs *P. aeruginosa* treated with ceftazidime), p<0.0001 (significance; *P. aeruginosa dsbA1* vs *P. aeruginosa* treated with ceftazidime), p<0.0001 (significance; *P. aeruginosa* treated with ceftazidime vs *P. aeruginosa dsbA1* treated with ceftazidime). For panels (A), (B), (D), and (E), the graphs show MIC values (µg/mL) from three biological experiments, each conducted as a single technical repeat; red dotted lines indicate the EUCAST clinical breakpoint for each antibiotic.

The online version of this article includes the following figure supplement(s) for figure 3:

**Figure supplement 1.** Assessment of off-target effects for clinical strains of *P. aeruginosa* and *S. maltophilia* that are deficient in oxidative protein folding.

aztreonam for *P. aeruginosa* CDC #769 (*Figure 3D*), and to representative compounds of all classes of anti-pseudomonal β-lactam drugs (piperacillin-tazobactam, aztreonam, and ceftazidime) for *P. aeruginosa* CDC #773 (*Figure 3E*). *P. aeruginosa* CDC #773 and its *dsbA1* mutant were further tested in a *G. mellonella* infection model using ceftazidime treatment (*Figure 3F*). In this case, no larvae survived 24 hr post-infection (*Figure 3F*; blue, light blue, and red survival curves), except for insects infected

with *P. aeruginosa* CDC #773 *dsbA1* and treated with ceftazidime at a final concentration below the EUCAST breakpoint, whereby 96.7% survival was recorded (*Figure 3F*; pink survival curves).

We have demonstrated the specific interaction of DsbA with the tested β-lactamase enzymes in our *E. coli* K-12 inducible system using gentamicin controls (*Figure 1* and *Supplementary file 2A*) and gene complementation (*Figure 1—figure supplement 1*). To confirm the specificity of this interaction in *P. aeruginosa*, we performed representative control experiments in one of our clinical strains, *P. aeruginosa* CDC #769. We first tested the general ability of *P. aeruginosa* CDC #769 *dsbA1* to resist antibiotic stress by recording MIC values against gentamicin and found it unchanged compared to its parent (*Figure 3—figure supplement 1A*). Gene complementation in clinical isolates is especially challenging and rarely attempted due to the high levels of resistance and lack of genetic tractability in these strains. Despite these challenges, to further ensure the specificity of the interaction of DsbA with tested β-lactamases in *P. aeruginosa*, we have complemented *dsbA1* from *P. aeruginosa* PAO1 into *P. aeruginosa* CDC #769 *dsbA1*. We found that complementation of *dsbA1* restores MICs to wild-type values for both tested β-lactam compounds (*Figure 3—figure supplement 1B*) further demonstrating that our results in *P. aeruginosa* clinical strains are not confounded by off-target effects.

Our data on the sensitization of AIM- and GES-expressing *P. aeruginosa* clinical isolates to commonly used anti-pseudomonal β-lactam drugs, combined with our previous results on strains producing β-lactamases from the OXA family (*Furniss et al., 2022*), show that our approach holds promise towards inactivating numerous clinically important *Pseudomonas*-specific enzymes. These include resistance determinants that cannot be currently targeted by classical β-lactamase inhibitor compounds (for example, enzymes from the OXA and AIM families *Tooke et al., 2019*) and, therefore, limit our treatment options.

**New treatment options for extremely drug-resistant *S. maltophilia* clinical isolates**. We have previously used our inducible *E. coli* K-12 experimental system to demonstrate that the function of the inhibitor-resistant class B3 metallo-β-lactamase L1-1 from *S. maltophilia* is dependent on DsbA (*Furniss et al., 2022*). By contrast, the second β-lactamase encoded on the chromosome of this species, L2-1, which we use as a negative control in this study (*Figures 1 and 2*), is not DsbA dependent. The hydrolytic spectra of these β-lactamases are exquisitely complementary (*Brooke, 2012*; *Calvopiña et al., 2017*), making this bacterium resistant to most β-lactam compounds commonly used for CF patients. Considering that L1 enzymes are the sole drivers of ceftazidime resistance, we wanted to investigate the DsbA dependency of L1-1 in its natural context to determine whether inhibition of oxidative protein folding potentiates the activity of complex cephalosporins against this pathogen.

We compromised disulfide bond formation in two clinical isolates of *S. maltophilia* (strains AMM and GUE; *Supplementary file 6, supplementary table 2*), by deleting the main *dsbA* gene cluster (directly adjacent *dsbA* and *dsbL* genes, with DsbL predicted to be a DsbA analogue *Landeta et al., 2018*) and recorded a drastic decrease of ceftazidime MIC values for both mutant strains (*Figure 4A and B*). Since *S. maltophilia* cannot be treated with ceftazidime, there is no EUCAST breakpoint available for this organism. That said, for both tested *dsbA dsbL* mutant strains, the recorded ceftazidime MIC values were lower than the ceftazidime EUCAST breakpoint for the related major pathogen *P. aeruginosa* (*Calza et al., 2003*).

In addition to being resistant to β-lactams, *S. maltophilia* is usually intrinsically resistant to colistin (*Amin et al., 2020*), which precludes the use of yet another broad class of antibiotics. Bioinformatic analysis on 106 complete *Stenotrophomonas* genomes revealed that most strains of this organism carry two chromosomally encoded MCR analogues that cluster with clinical MCR-5 and MCR-8 proteins (*Supplementary file 4*). We have previously found the activity of all clinical MCR enzymes to be dependent on the presence of DsbA (*Furniss et al., 2022*), thus we compared the colistin MIC value of the *S. maltophilia* AMM *dsbA dsbL* strain to that of its parent. We found that impairment of disulfide bond formation in this strain resulted in a decrease of its colistin MIC value from 32 µg/mL to 0.75 µg/mL (*Figure 4C*). Once more, there is no colistin EUCAST breakpoint available for *S. maltophilia*, but a comparison with the colistin breakpoint for *P. aeruginosa* (4 µg/mL) demonstrates the magnitude of the effects that we observe.

Since the *dsbA* and *dsbL* are organized in a gene cluster in *S. maltophilia*, we wanted to ensure that our results reported above were exclusively due to disruption of disulfide bond formation in this organism. First, we recorded gentamicin MIC values for *S. maltophilia* AMM *dsbA dsbL* and found them to be unchanged compared to the gentamicin MICs of the parent strain (*Figure 3—figure*

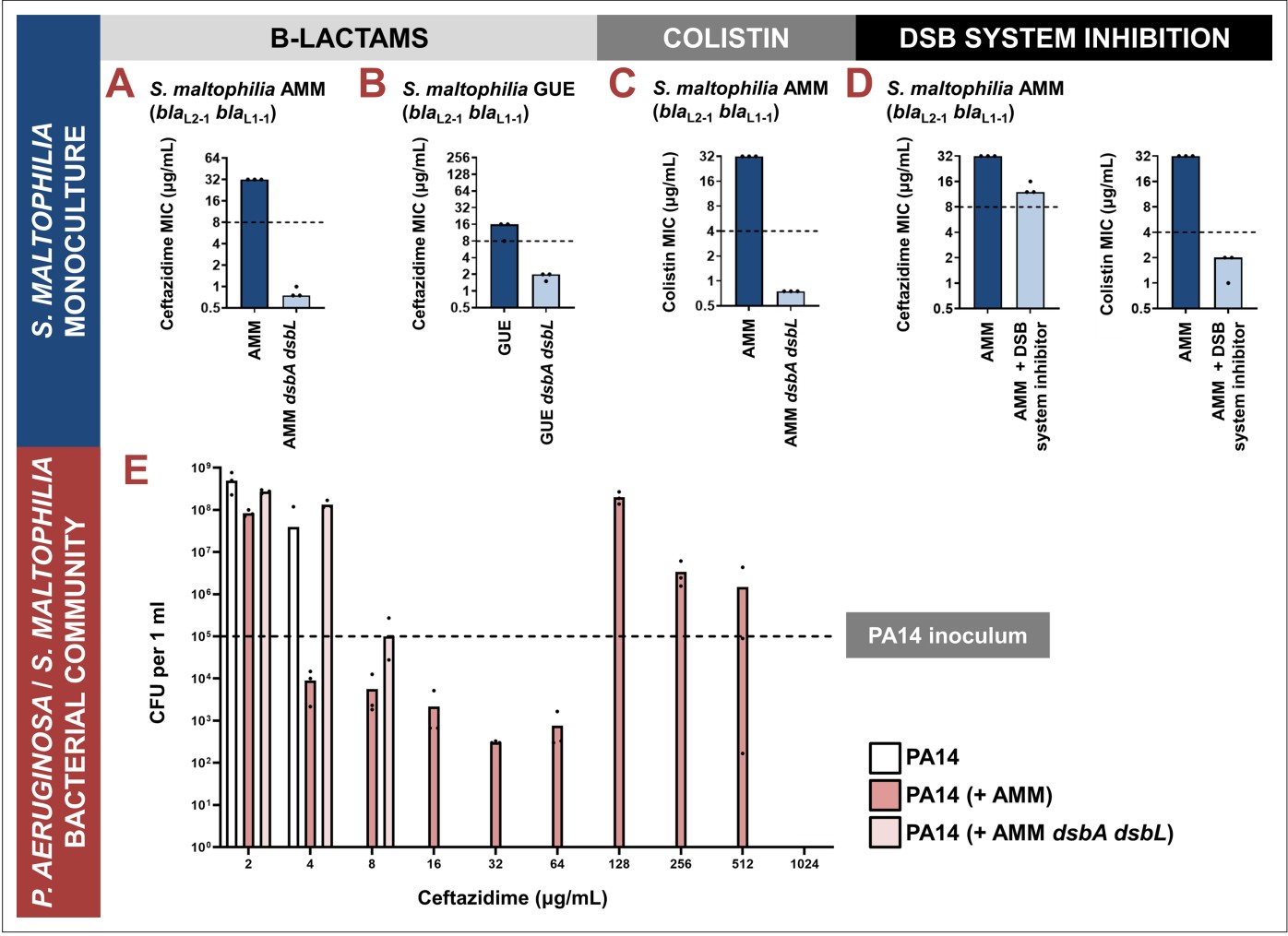

**Figure 4.** Impairment of disulfide bond formation allows treatment of Stenotrophomonas maltophilia with β-lactam and colistin antibiotics while also inhibiting cross-protection between *S. maltophilia* and *Pseudomonas aeruginosa* in mixed communities. (**A–D**) Impairment of disulfide bond formation allows the treatment of *Stenotrophomonas maltophilia* clinical strains with β-lactam and colistin antibiotics. (**A, B**) Deletion of *dsbA dsbL* in the *S. maltophilia* AMM and *S. maltophilia* GUE clinical isolates results in drastic decrease of their ceftazidime MIC values. (**C**) Deletion of *dsbA dsbL* in the *S. maltophilia* AMM clinical strain results in a drastic decrease of its colistin MIC value. (**D**) Use of a small-molecule inhibitor of DsbB against the *S. maltophilia* AMM clinical strain results in a decrease of its ceftazidime and colistin MIC values. For panels (**A–D**), graphs show MIC values (μg/mL) from three biological experiments; for β-lactam MIC assays, each experiment was conducted as a single technical repeat, whereas for colistin MIC assays, each experiment was conducted in technical triplicate. In the absence of EUCAST clinical breakpoints for *S. maltophilia*, the black dotted lines indicate the EUCAST clinical breakpoint for each antibiotic for the related pathogen *P. aeruginosa*. (**E**) Protection of *P. aeruginosa* by *S. maltophilia* clinical strains is dependent on oxidative protein folding. The susceptible *P. aeruginosa* strain PA14 can survive exposure to ceftazidime up to a maximum concentration of 4 μg/mL when cultured in isolation (white bars). By contrast, if co-cultured in the presence of *S. maltophilia* AMM, which can hydrolyze ceftazidime through the action of its L1-1 β-lactamase enzyme, *P. aeruginosa* PA14 can survive and actively grow in concentrations of ceftazidime as high as 512 μg/mL (dark pink bars). This protection is abolished if *P. aeruginosa* PA14 is co-cultured with *S. maltophilia* AMM *dsbA dsbL* (light pink bars), where L1-1 is inactive (as shown in (A) and **Furniss et al., 2022**). The graph shows *P. aeruginosa* PA14 colony-forming unit (CFU) counts for each condition; three biological replicates were conducted in technical triplicate, and mean CFU values are shown. The black dotted line indicates the *P. aeruginosa* PA14 inoculum. The mean CFU values used to generate this figure are presented in **Supplementary file 2B**.

The online version of this article includes the following figure supplement(s) for figure 4:

**Figure supplement 1.** Protection of *P. aeruginosa* by *S. maltophilia* clinical strains is dependent on oxidative protein folding.

supplement 1C). This confirms that disruption of disulfide bond formation does not compromise the general ability of this organism to resist antibiotic stress. Next, we complemented *S. maltophilia* AMM *dsbA dsbL*. The specific oxidative roles and exact regulation of DsbA and DsbL in *S. malto-philia* remain unknown. For this reason and considering that genetic manipulation of extremely drug-resistant organisms is challenging, we used our genetic construct optimized for complementing *P.*

*aeruginosa* CDC #769 *dsbA1* with *dsbA1* from *P. aeruginosa* PAO1 (*Figure 3—figure supplement 1B*) to also complement *S. maltophilia* AMM *dsbA dsbL*. We based this approach on the fact that DsbA proteins from one species have been commonly shown to be functional in other species (*Ng et al., 1997*; *Paxman et al., 2009*; *Santos-Martin et al., 2021*; *Urban et al., 2001*). Indeed, we found that complementation of *S. maltophilia* AMM *dsbA dsbL* with *dsbA1* from *P. aeruginosa* PAO1 restores MICs to wild-type values for both ceftazidime and colistin (*Figure 3—figure supplement 1D*), conclusively demonstrating that our results in *S. maltophilia* are not confounded by off-target effects.

The DSB proteins have been shown to play a central role in bacterial virulence, and in this context, they have been proposed as promising targets against bacterial pathogenesis (*Heras et al., 2009*; *Landeta et al., 2018*; *Heras et al., 2015*). As a result, several laboratory compounds against both DsbA (*Duprez et al., 2015*; *Totsika et al., 2018*) and its partner protein DsbB (*Landeta et al., 2015*), which maintains DsbA in a catalytically active state (*Kadokura et al., 2003*), have been developed. We have successfully used one of these inhibitors, 4,5-dichloro-2-(2-chlorobenzyl)pyridazin-3-one, termed 'compound 12' in *Yong et al., 2012*, to achieve sensitization of clinical strains of Enterobacteria to β-lactam and colistin antibiotics (*Furniss et al., 2022*). Here, we used a derivative compound, 4,5-dibromo-2-(2-chlorobenzyl)pyridazin-3(2 H)-one, termed 'compound 36' in *Landeta et al., 2017*, which is an improved analog of compound 12 and has been shown to target several DsbB proteins from Gram-negative pathogens that share 20–80% in protein identity. Compound 36 was previously shown to inhibit disulfide bond formation in *P. aeruginosa* via covalently binding onto one of the four essential cysteine residues of DsbB in the DsbA-DsbB complex (*Landeta et al., 2017*). Since *S. maltophilia* DsbB shares ~28% protein sequence identity with analogues from *P. aeruginosa*, we reasoned that this pathogen could be a good candidate for testing DSB system inhibition. Exposure of *S. maltophilia* AMM to the DSB inhibitor lowered its ceftazidime MIC value by at least 16–20 µg/mL and decreased its colistin MIC value from 32 µg/mL to 2 µg/mL (*Figure 4D*); this decrease in the colistin MIC is commensurate with the results we obtained for the *S. maltophilia* AMM *dsbA dsbL* strain (*Figure 4C*). The activity of compound 36 is specific to inhibition of disulfide bond formation since the gentamicin MIC values of *S. maltophilia* AMM remain unchanged in the presence of the inhibitor and treatment of *S. maltophilia* AMM *dsbA dsbL* with the compound does not affect its colistin MIC value (*Figure 3—figure supplement 1E*). Considering that this inhibitor has not been specifically optimized for *S. maltophilia* strains, the recorded drops in MIC values (*Figure 4D*) are encouraging and suggest that the DSB system proteins are tractable targets against species-specific resistance determinants in this pathogen.

Currently, the best clinical strategy against *S. maltophilia* is to reduce the likelihood of infection (*Gibb and Wong, 2021*); therefore, novel treatment strategies against this organism are desperately needed. Overall, our results on targeting oxidative protein folding in this organism show promise for the generation of therapeutic avenues that are compatible with mainstream antibiotics (β-lactams and polymyxins), which are commonly used for the treatment of other pathogens, for example *P. aeruginosa*, in CF lung infections.

**Inhibition of cross-resistance in *S. maltophilia* - *P. aeruginosa* mixed communities**. The antibiotic resistance mechanisms of *S. maltophilia* impact the antibiotic tolerance profiles of other organisms that are found in the same infection environment. *S. maltophilia* hydrolyzes all β-lactam drugs through the action of its L1 and L2 β-lactamases (*Brooke, 2012*; *Calvopiña et al., 2017*). In doing so, it has been experimentally shown to protect other pathogens that are, in principle, susceptible to treatment, such as *P. aeruginosa* (*Bottery et al., 2022*). This protection, in turn, allows active growth of otherwise treatable *P. aeruginosa* in the presence of complex β-lactams, like imipenem (*Bottery et al., 2022*), and, at least in some conditions, increases the rate of resistance evolution of *P. aeruginosa* against these antibiotics (*Quinn et al., 2022*).

We wanted to investigate whether our approach would be useful in abrogating interspecies interactions that are relevant to CF infections. We posited that ceftazidime resistance in *S. maltophilia* is largely driven by L1-1, an enzyme that we can incapacitate by targeting disulfide bond formation (*Furniss et al., 2022*; *Figure 4A, B and D*). As such, impairment of oxidative protein folding in *S. maltophilia* should allow treatment of this organism with ceftazidime, and at the same time eliminate any protective effects that benefit susceptible strains of co-occurring organisms. With ceftazidime being a standard anti-pseudomonal drug, and in view of the interactions reported between *P. aeruginosa* and *S. maltophilia* (*Bottery et al., 2022*; *Quinn et al., 2022*; *Law et al., 2022*), we chose to test

this hypothesis using *S. maltophilia* AMM and a *P. aeruginosa* strain that is sensitive to β-lactam antibiotics, *P. aeruginosa* PA14. We followed established co-culture protocols for these organisms (*Bottery et al., 2022*) and first monitored the survival and growth of *P. aeruginosa* under ceftazidime pressure in monoculture, or in the presence of *S. maltophilia* strains. Due to the naturally different growth rates of these two species (*S. maltophilia* grows much slower than *P. aeruginosa*), especially in laboratory conditions, the protocol we followed (*Bottery et al., 2022*) requires *S. maltophilia* to be grown for 6 hr prior to co-culturing it with *P. aeruginosa*. To ensure that at this point in the experiment, our two *S. maltophilia* strains, with and without *dsbA*, had grown comparatively to each other, we determined their cell densities (*Figure 4—figure supplement 1A*). We found that *S. maltophilia* AMM *dsbA dsbL* had grown at a similar level as the wild-type strain, and both were at a higher cell density (~$10^7$ colony forming units (CFUs)) compared to the *P. aeruginosa* PA14 inoculum ($5 \times 10^4$ CFUs).

*P. aeruginosa* PA14 monoculture cannot grow in the presence of more than 4 µg/mL of ceftazidime (*Figure 4E*; white bars). However, the same strain can actively grow in concentrations of ceftazidime up to 512 µg/mL in the presence of *S. maltophilia* AMM (*Figure 4E*; dark pink bars), showing that the protective effects previously observed with imipenem (*Bottery et al., 2022*) are applicable to other clinically relevant β-lactam antibiotics. Cross-resistance effects are most striking at concentrations of ceftazidime above 64 µg/ml; for amounts between 16 and 64 µg/ml, *P. aeruginosa* survives in the presence of *S. maltophilia*, but does not actively grow. This is in agreement with previous observations showing that the expression of L1-1 is induced by the presence of complex β-lactams (*Okazaki and Avison, 2008*). In this case, the likely increased expression of L1-1 in *S. maltophilia* grown in concentrations of ceftazidime equal to or higher than 128 µg/ml promotes ceftazidime hydrolysis and decrease of the active antibiotic concentration, in turn, shielding the susceptible *P. aeruginosa* strain. By contrast, protective effects are almost entirely absent when *P. aeruginosa* PA14 is co-cultured with *S. maltophilia* AMM *dsbA dsbL*, which cannot hydrolyze ceftazidime efficiently because L1-1 activity is impaired (*Furniss et al., 2022*; *Figure 4A, B and D*). In fact, in these conditions, *P. aeruginosa* PA14 only survives in concentrations of ceftazidime up to 8 µg/mL (*Figure 4E*; light pink bars), 64-fold lower than what it can endure in the presence of *S. maltophilia* AMM (*Figure 4E*; dark pink bars).

To ensure that ceftazidime treatment leads to eradication of both *P. aeruginosa* and *S. maltophilia* when disulfide bond formation is impaired in *S. maltophilia*, we monitored the abundance of both strains in each synthetic community for select antibiotic concentrations (*Figure 4—figure supplement 1B*). In this experiment, we largely observed the same trends as in *Figure 4E*. At low antibiotic concentrations, for example 4 µg/mL of ceftazidime, *S. maltophilia* AMM is fully resistant and thrives, thus outcompeting *P. aeruginosa* PA14 (dark pink and dark blue bars in *Figure 4—figure supplement 1B*). The same can also be seen in *Figure 4E*, whereby decreased *P. aeruginosa* PA14 CFUs are recorded. By contrast, *S. maltophilia* AMM *dsbA dsbL* already displays decreased growth at 4 µg/mL of ceftazidime because of its non-functional L1-1 enzyme, allowing comparatively higher growth of *P. aeruginosa* (light pink and light blue bars in *Figure 4—figure supplement 1B*). Despite the competition between the two strains, *P. aeruginosa* PA14 benefits from *S. maltophilia* AMM's high hydrolytic activity against ceftazidime, which allows it to survive and grow in high antibiotic concentrations even though it is not resistant (see 128 µg/mL; dark pink and dark blue bars in *Figure 4—figure supplement 1B*). In stark opposition, without its disulfide bond in *S. maltophilia* AMM *dsbA dsbL*, L1-1 cannot confer resistance to ceftazidime, resulting in killing of *S. maltophilia* AMM *dsbA dsbL* and, consequently, also of *P. aeruginosa* PA14 (see 128 µg/mL; light pink and light blue bars in *Figure 4—figure supplement 1B*).

The data presented here show that, at least under laboratory conditions, targeting protein homeostasis pathways in specific recalcitrant pathogens has the potential not only to alter their own antibiotic resistance profiles (*Figures 3 and 4*), but also to influence the antibiotic susceptibility profiles of other bacteria that co-occur in the same conditions (*Figure 5*). Admittedly, the conditions in a living host are too complex to draw direct conclusions from this experiment. That said, our results show promise for infections, where pathogen interactions affect treatment outcomes, and whereby their inhibition might facilitate treatment.

## Discussion

Impairment of cell envelope protein homeostasis through interruption of disulfide bond formation has potential as a broad-acting strategy against AMR in Gram-negative bacteria (*Furniss et al., 2022*). Here, we focus on the benefits of such an approach against pathogens encountered in challenging

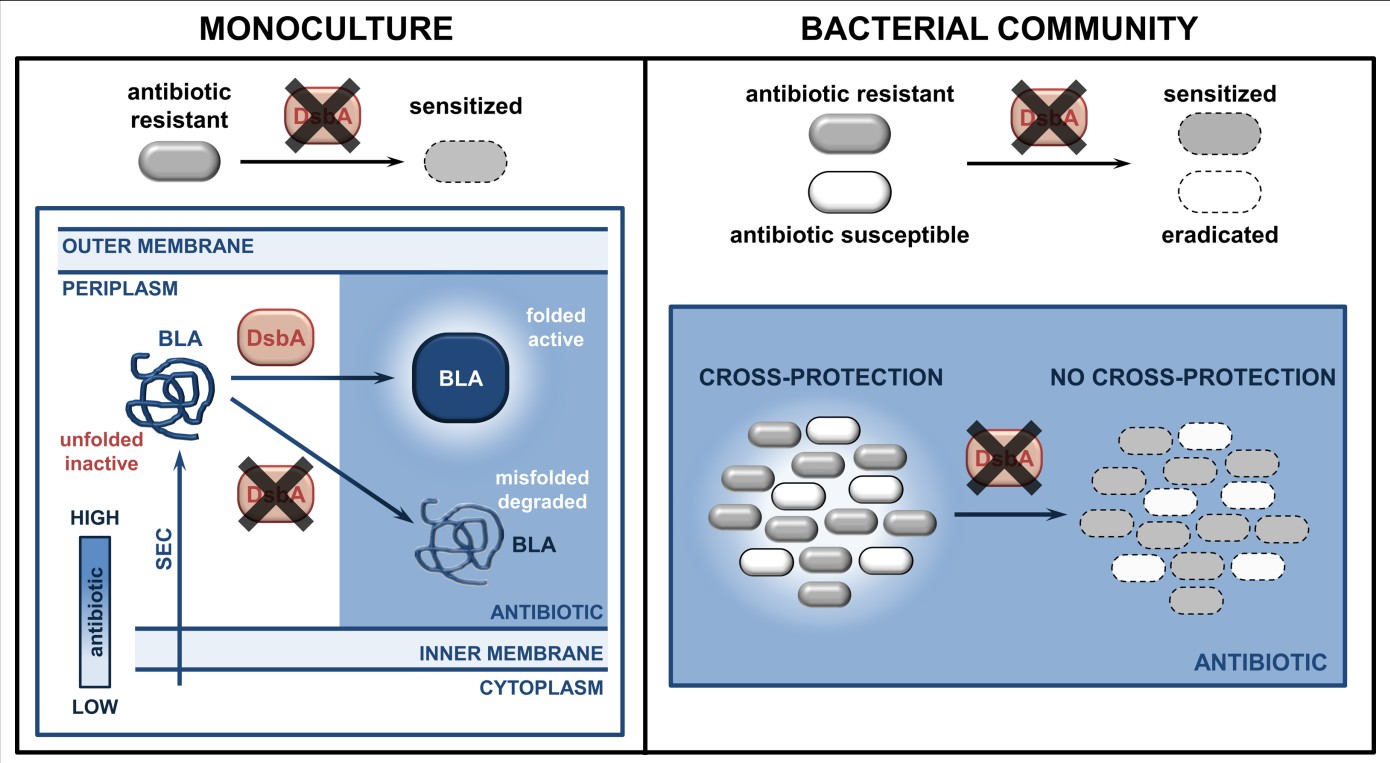

**Figure 5.** Inhibition of oxidative protein folding counters antibiotic resistance and inter-species interactions in CF-associated pathogens. (**Left**) After Sec translocation to the periplasm and DsbA-assisted folding, cysteine-containing species-specific β-lactamase enzymes from recalcitrant pathogens, like *P. aeruginosa* or *S. maltophilia*, are active and can hydrolyze β-lactam antibiotics. However, in the absence of their disulfide bonds, DsbA-dependent β-lactamases either degrade or misfold, and thus can no longer confer resistance to β-lactam compounds. (**Right**) In multispecies bacterial communities, bacteria that degrade antibiotics, for example species producing β-lactamases, can protect antibiotic-susceptible strains. Targeting disulfide bond formation impairs interbacterial interactions that are reliant on the activity of DsbA-dependent β-lactamase enzymes, allowing eradication of both bacterial species.

The online version of this article includes the following source data and figure supplement(s) for figure 5:

**Figure supplement 1.** The activity of additional species-specific β-lactamases depends on disulfide bond formation.

**Figure supplement 1—source data 1.** Original files of the full raw unedited immunoblots used to prepare *Figure 5—figure supplement 1B*.

**Figure supplement 1—source data 2.** Uncropped immunoblots used to prepare *Figure 5—figure supplement 1B*.

**Figure supplement 2.** Complementation of *dsbA* restores the β-lactam MIC values for *E. coli* MC1000 *dsbA* expressing β-lactamases.

infection settings by studying organisms found in the CF lung. In particular, we show that incapacitation of oxidative protein folding compromises the function of diverse β-lactamases that are specific to CF-associated bacteria, like *P. aeruginosa* and *Burkholderia* complex (*Figures 1 and 2*). Furthermore, we find that the effects we observe at the enzyme level are applicable to multiple MDR *P. aeruginosa* and extremely drug-resistant *S. maltophilia* clinical strains, both *in vitro* (*Figures 3A, B, D, E and 4A, B*) and in an *in vivo* model of infection (*Figure 3C and F*). Our findings, so far, concern β-lactamases encoded by enteric pathogens (discussed in *Furniss et al., 2022*) or CF-associated organisms (discussed in *Furniss et al., 2022* and in this study). Nonetheless, many other environmental bacteria are opportunistic human pathogens and encode β-lactamase genes that make them highly resistant to antibiotic treatment (*Brooke, 2012*; *Nakae et al., 1999*; *Wiersinga et al., 2006*). The ubiquitous nature of disulfide bond formation systems across Gram-negative species guarantees that the same approach can be expanded. To provide some proof on this front, we investigated two additional class B3 metallo-β-lactamases, POM-1 produced by *Pseudomonas otitidis* and SMB-1 encoded on the chromosome of *Serratia marcescens* (*Supplementary file 6, supplementary table 1*). We tested these enzymes in our inducible *E. coli* K-12 system and found that their activities are indeed DsbA dependent (*Figure 5—figure supplement 1A*, *Figure 5—figure supplement 2*, and *Supplementary file 2A*), with SMB-1 degrading in the absence of DsbA and POM-1 suffering a folding defect

(*Figure 5—figure supplement 1B and C*). Since 57% of β-lactamase phylogenetic families that are found in pathogens and organisms capable of causing opportunistic infections contain members with two or more cysteines (*Supplementary file 1*), we expect that thousands of enzymes rely on DsbA for their stability and function. Focusing solely on the β-lactamase families that we have investigated here and previously (*Furniss et al., 2022*; 17 phylogenetic families), we estimate that upwards of 575 discrete proteins are DsbA dependent. This encompasses enzymes specific to pathogens with very limited treatment options, for example the *Burkholderia* complex (*Figure 1* and *Supplementary file 2A*) and *S. maltophilia* (*Figure 4A, B and D*), as well as 145 β-lactamases that cannot be inhibited by classical adjuvant approaches, like class B enzymes (*Tooke et al., 2019*) from the AIM, L1, POM, and SMB families (*Figures 1 and 3A–C*, *Figure 5—figure supplement 1*, and *Supplementary file 2A*).

Of the organisms studied in this work, *S. maltophilia* deserves further discussion because of its unique intrinsic resistance profile. The prognosis of CF patients with *S. maltophilia* lung carriage is still debated (*Widder et al., 2022*; *Barsky et al., 2017*; *Berdah et al., 2018*; *Cogen et al., 2015*; *Denton et al., 1996*; *Goss et al., 2004*; *Goss et al., 2002*; *Waters et al., 2011*; *Watson et al., 2022*), largely because studies with extensive and well-controlled patient cohorts are lacking. This notwithstanding, the therapeutic options against this pathogen are currently limited to one non-β-lactam antibiotic-adjuvant combination, which is not always effective, trimethoprim-sulfamethoxazole (*Al-Jasser, 2006*; *Hu et al., 2011*; *Mojica et al., 2022*; *Toleman et al., 2007*), and a few last-line β-lactam drugs, like the fifth-generation cephalosporin cefiderocol and the combination aztreonam-avibactam. Resistance to commonly used antibiotics causes many problems during treatment and, as a result, infections that harbor *S. maltophilia* have high case fatality rates (*Brooke, 2012*). This is not limited to CF patients, as *S. maltophilia* is a major cause of death in children with bacteremia (*Alsuhaibani et al., 2021*). We find that targeting disulfide bond formation in this species allows its treatment with cephalosporins, like ceftazidime (*Figure 4A, B and D*) and, at the same time, leads to colistin potentiation (*Figure 4C and D*). Our results create a foundation for extending the usability of two invaluable broad-acting antibiotic classes against this challenging organism. At the same time, *S. maltophilia* is often found to co-exist in the CF lung with other pathogens like *P. aeruginosa* (*Widder et al., 2022*; *de Vrankrijker et al., 2010*; *Terlizzi et al., 2023*). Even though current studies are confined to laboratory settings (*Bottery et al., 2022*), it is likely that interactions between these two species make treatment of polymicrobial infections more complex. Here, we demonstrate that by compromising L1-1 through impairing protein homeostasis in *S. maltophilia* (*Figure 4A, B and D* and *Furniss et al., 2022*), in addition to generating new treatment options (*Figure 4A–D*), we abolish the capacity of this organism to protect other species (*Figure 4E*). Since similar bacterial interactions are documented in resistant infections (*Bottery et al., 2021*), it can be expected that our approach will yield analogous results for other co-existing CF lung pathogens that produce DsbA-dependent β-lactamases (*Furniss et al., 2022*), for example *P. aeruginosa* and *S. aureus* (*Fazli et al., 2009*; *Korgaonkar et al., 2013*) or *K. pneumoniae* and *Acinetobacter baumannii* (*Semenec et al., 2023*; *Figure 5*).

More generally, our findings serve as proof of principle of the added benefits of strategies that aim to incapacitate resistance determinants like β-lactamases. These proteins threaten the most widely prescribed class of antibiotics worldwide (*Versporten et al., 2018*) and, at the same time, can promote cross-resistance between pathogens found in polymicrobial infections. It is therefore important to continue developing β-lactamase inhibitors, which, so far, have been one of the biggest successes in our battle against AMR (*Tooke et al., 2019*; *Laws et al., 2019*). That said, the deployment of broad-acting small molecules with the capacity to bind and effectively inhibit thousands of clinically important β-lactamases (7741 distinct documented enzymes *Naas et al., 2017*; *Supplementary file 1*) is challenging and, eventually, leads to the emergence of β-lactamase variants that are resistant to combination therapy. As such, the development of additional alternative strategies that can broadly incapacitate these resistance proteins, ideally without the need to bind to their active sites, is critical. This has been shown to be possible through metal chelation for class B metallo-β-lactamases (*Sun et al., 2020*). Adding to this, our previous work (*Furniss et al., 2022*) and the results presented here lay the groundwork for exploiting accessible cell envelope proteostasis processes to generate new resistance breakers. Inhibiting such systems has untapped potential for the design of broad-acting next-generation therapeutics, which simultaneously compromise multiple resistance mechanisms (*Furniss et al., 2022*), and also for the development of species- or infection-specific approaches that are well suited for the treatment of complex polymicrobial communities (*Figure 5*).

# Materials and methods

## Reagents and bacterial growth conditions

Unless otherwise stated, chemicals and reagents were acquired from Sigma-Aldrich or Thermo Fisher Scientific, growth media were purchased from Oxoid and antibiotics were obtained from Melford Laboratories. Lysogeny broth (LB) (10 g/L NaCl) and agar (1.5% w/v) were used for routine growth of all organisms at 37 °C with shaking at 220 RPM, as appropriate. Mueller-Hinton (MH) broth and agar (1.5% w/v) were used for Minimum Inhibitory Concentration (MIC) assays. Growth media were supplemented with the following, as required: 0.25 mM Isopropyl β-D-1-thiogalactopyranoside (IPTG), 50 µg/mL kanamycin, 100 µg/mL ampicillin, 33 µg/mL chloramphenicol, 33 µg/mL gentamicin (for cloning purposes), 400–600 µg/mL gentamicin (for genetic manipulation of *P. aeruginosa* and *S. maltophilia* clinical isolates), 12.5 µg/mL tetracycline (for cloning purposes), 100–400 µg/mL tetracycline (for genetic manipulation of *P. aeruginosa* clinical isolates), 50 µg/mL streptomycin (for cloning purposes), 2000–5000 µg/mL streptomycin (for genetic manipulation of *P. aeruginosa* clinical isolates), and 6000 µg/mL streptomycin (for genetic manipulation of *S. maltophilia* clinical isolates).

## Construction of plasmids and bacterial strains

Bacterial strains, plasmids, and oligonucleotides used in this study are listed in *Supplementary file 6, supplementary tables 2–4*, respectively. DNA manipulations were conducted using standard methods. KOD Hot Start DNA polymerase (Merck) was used for all PCR reactions according to the manufacturer's instructions, oligonucleotides were synthesized by Sigma-Aldrich and restriction enzymes were purchased from New England Biolabs. All constructs were DNA sequenced and confirmed to be correct before use.

Genes for β-lactamase enzymes were amplified from genomic DNA extracted from clinical isolates (*Supplementary file 6, supplementary table 5*) with the exception of *bps-1m, bps-6, carb-2, ftu-1*, and *smb-1*, which were synthesized by GeneArt Gene Synthesis (Thermo Fisher Scientific). β-lactamase genes were cloned into the IPTG-inducible plasmid pDM1 using primers P1-P16. All StrepII-tag fusions of β-lactamase enzymes (constructed using primers P3, P5, P7, P9, P11, P13, P15, and P17-23) have a C-terminal StrepII tag (GSAWSHPQFEK).

*P. aeruginosa dsbA1* mutants and *S. maltophilia dsbA dsbL* mutants were constructed by allelic exchange, as previously described (*Vasseur et al., 2005*). Briefly, the *dsbA1* gene area of *P. aeruginosa* strains (including the *dsbA1* gene and ~600 bp on either side of this gene) was amplified (primers P24/P25) and the obtained DNA was sequenced to allow for accurate primer design for the ensuing cloning step. The pKNG101-*dsbA1* plasmid was then used for deletion of the *dsbA1* gene in *P. aeruginosa* G4R7 and *P. aeruginosa* G4R7, as before (*Furniss et al., 2022*). For the deletion of *dsbA1* in *P. aeruginosa* CDC #769 and *P. aeruginosa* CDC #773, ~500 bp DNA fragments upstream and downstream of the *dsbA1* gene were amplified using *P. aeruginosa* CDC #769 or *P. aeruginosa* CDC #773 genomic DNA (primers P28/P29 (upstream) and P30/P31 (downstream)). Fragments containing both regions were then obtained by overlapping PCR (primers P28/P31) and inserted into the XbaI/BamHI sites of pKNG102, resulting in plasmids pKNG102-*dsbA1*-769 and pKNG102-*dsbA1*-773. For *S. maltophilia* strains, the *dsbA dsbL* gene area (including the *dsbA dsbL* genes and ~1000 bp on either side of these genes) was amplified (primers P26/P27), and the obtained DNA was sequenced to allow for accurate primer design for the ensuing cloning step. Subsequently, ~700 bp DNA fragments upstream and downstream of the *dsbA dsbL* genes were amplified using *S. maltophilia* AMM or *S. maltophilia* GUE genomic DNA (primers P32/P33 (upstream) and P34/P35 (downstream)). Fragments containing both of these regions were then obtained by overlapping PCR (primers P32/35) and inserted into the XbaI/BamHI sites of pKNG101, resulting in plasmids pKNG101-*dsbA dsbL*-AMM and pKNG101-*dsbA dsbL*-GUE. The suicide vector pKNG101 (*Kaniga et al., 1991*) and its derivative pKNG102 are not replicative in *P. aeruginosa* or *S. maltophilia*; both vectors are maintained in *E. coli* CC118 λ pir and mobilized into *P. aeruginosa* and *S. maltophilia* strains by triparental conjugation. For *P. aeruginosa,* integrants were selected on Vogel Bonner Minimal medium supplemented with streptomycin (for *P. aeruginosa* G4R7 and *P. aeruginosa* G6R7) or tetracycline (for *P. aeruginosa* CDC #769 and *P. aeruginosa* CDC #773). For *S. maltophilia,* integrants were selected on MH agar supplemented with streptomycin and ampicillin. Successful integrants were confirmed using PCR, and mutants were resolved by exposure to 20% sucrose. Gene deletions were confirmed via colony PCR and DNA sequencing (primers P24/P25).

*P. aeruginosa* PA14, *S. maltophilia* AMM, and *S. maltophilia* AMM *dsbA dsbL* were labeled with a gentamicin resistance marker using mini-Tn*7* delivery transposon-based vectors adapted from *Zobel et al., 2015*. The non-replicative vectors pTn7-M (labeling with gentamicin resistance only (*accC* gene), for *P. aeruginosa* PA14) and pBG42 (labeling with gentamicin resistance and msfGFP, for *S. maltophilia* strains) were mobilized into the respective recipients using conjugation, in the presence of a pTNS2 plasmid expressing the TnsABC + D-specific transposition pathway. Correct insertion of the transposon into the *att*Tn*7* site was confirmed via colony PCR and DNA sequencing (primers P44/P45 for *P. aeruginosa*, primers P46/P47 for *S. maltophilia*).

*P. aeruginosa* CDC #769 *dsbA1* and *S. maltophilia* AMM *dsbA dsbL* were complemented with DsbA1 from *P. aeruginosa* PAO1 using a mini-Tn*7* delivery transposon-based vector adapted from *Zobel et al., 2015*. Briefly, the *msfGFP* gene of pBG42 was replaced with the *dsbA1* gene of *P. aeruginosa* PAO1 by HiFi DNA assembly according to the manufacturer's instructions (NEBuilder HiFi DNA Assembly, New England Biolabs). The vector was linearized with primers P36/P37 and the *dsbA1* gene of *P. aeruginosa* PAO1 was amplified from genomic DNA using primers P38/P39. *msfGFP* amplified from pBG42 with primers P40/P41 was reintroduced onto the vector under the PEM7 promoter between the HindIII and BamHI sites of pBG42 (*García-Gutiérrez et al., 2020*) resulting in plasmid pBG42-PAO1*dsbA1*. Correct assembly of pBG42-PAO1*dsbA1* was confirmed by colony PCR (primers P42/P43) and DNA sequencing. pBG42-PAO1*dsbA1* was mobilized into the recipient strains using conjugation, in the presence of a pTNS2 plasmid expressing the TnsABC + D-specific transposition pathway. GFP positive colonies were screened using colony PCR and correct insertion of the transposon into the *att*Tn*7* site of clinical strains was confirmed via DNA sequencing (primers P44/P45 for *P. aeruginosa*, primers P46/P47 for *S. maltophilia*).

Due to the evolving nomenclature and classification of L1 and L2 enzymes at the time of this study's publication, the designations L1-1 and L2-1 are used here in a generic sense to refer to enzymes belonging to the L1 and L2 β-lactamase families that are variants of L1-1 and L2-1, respectively.

## Minimum inhibitory concentration (MIC) assays

Unless otherwise stated, antibiotic MIC assays were carried out in accordance with the EUCAST recommendations (*Kadeřábková et al., 2024*) using ETEST strips (BioMérieux). Briefly, overnight cultures of each strain to be tested were standardized to $OD_{600}$ 0.063 in 0.85% NaCl (equivalent to McFarland standard 0.5) and distributed evenly across the surface of MH agar plates. ETEST strips were placed on the surface of the plates, evenly spaced, and the plates were incubated for 18–24 hr at 37 °C. MICs were read according to the manufacturer's instructions. MICs were also determined using the Broth Microdilution (BMD) method in accordance with the EUCAST recommendations (*Kadeřábková et al., 2024*) for specific β-lactams, as required, and for colistin sulfate (Acros Organics). Briefly, a series of antibiotic concentrations was prepared by twofold serial dilution in MH broth in a clear-bottomed 96-well microtiter plate (Corning). The strain to be tested was added to the wells at approximately $5 \times 10^5$ CFUs per well, and plates were incubated for 18–24 hr at 37 °C. The MIC was defined as the lowest antibiotic concentration with no visible bacterial growth in the wells. When used for MIC assays, tazobactam was included at a fixed concentration of 4 µg/mL, in accordance with the EUCAST guidelines. All *S. maltophilia* MICs were performed in synthetic CF sputum medium (SCFM) as described in *Palmer et al., 2007*, using ETEST strips (for β-lactam antibiotics) or the BMD method (for colistin). For *S. maltophilia* GUE, imipenem at a final concentration of 5 µg/mL was added to the overnight cultures to induce β-lactamase production.

The covalent DsbB inhibitor 4,5-dibromo-2-(2-chlorobenzyl)pyridazin-3(2 H)-one (*Landeta et al., 2017*) was used to chemically impair the function of the DSB system in *S. maltophilia* strains. Inactivation of DsbB results in abrogation of DsbA function (*Kishigami et al., 1995*) only in media free of small-molecule oxidants (*Dailey and Berg, 1993*). Therefore, MIC assays involving chemical inhibition of the DSB system were performed using SCFM media prepared as described in *Palmer et al., 2007*, except that L-cysteine was omitted. Either DMSO (vehicle control) or the covalent DsbB inhibitor 4,5-dibromo-2-(2-chlorobenzyl)pyridazin-3(2 H)-one (*Landeta et al., 2017*; Bioduro-Sundia; [1]H-NMR and LCMS spectra are provided in *Supplementary file 5*), at a final concentration of 50 µM, were added to the cysteine-free SCFM medium, as required.

## SDS-PAGE analysis and immunoblotting

Samples for immunoblotting were prepared as follows. Strains to be tested were grown on LB agar plates as lawns in the same manner as for MIC assays described above. Bacteria were collected using an inoculating loop and resuspended in LB to $OD_{600}$ 2.0. The cell suspensions were centrifuged at 10,000 $x$ $g$ for 10 min and bacterial pellets were lysed by addition of BugBuster Master Mix (Merck Millipore) for 25 min at room temperature with gentle agitation. Subsequently, lysates were centrifuged at 10,000 $x$ $g$ for 10 min at 4 °C and the supernatant was added to 4 x Laemmli buffer. Samples were boiled for 5 min before separation by SDS-PAGE.

SDS-PAGE analysis was carried out using 10% BisTris NuPAGE gels (Thermo Fisher Scientific) and MES/SDS running buffer prepared according to the manufacturer's instructions; pre-stained protein markers (SeeBlue Plus 2, Thermo Fisher Scientific) were included. Proteins were transferred to Amersham Protran nitrocellulose membranes (0.45 µm pore size, GE Life Sciences) using a Trans-Blot Turbo transfer system (Bio-Rad) before blocking in 3% w/v Bovine Serum Albumin (BSA)/TBS-T (0.1 % v/v Tween 20) or 5% w/v skimmed milk/TBS-T and addition of primary and secondary antibodies. The following primary antibodies were used in this study: Strep-Tactin-AP conjugate (Iba Lifesciences; dilution 1:3000 in 3 w/v % BSA/TBS-T), and mouse anti-DnaK 8E2/2 antibody (Enzo Life Sciences; dilution 1:10,000 in 5% w/v skimmed milk/TBS-T). Goat anti-mouse IgG-AP conjugate (Sigma-Aldrich; dilution 1:6000 in 5% w/v skimmed milk/TBS-T) was used as a secondary antibody in this study. Membranes were washed three times for 5 min with TBS-T prior to development. Development for AP conjugates was carried out using SigmaFast BCIP/NBT tablets.

Immunoblot samples were also analyzed for total protein content. SDS-PAGE analysis was carried out using 10% BisTris NuPAGE gels (Thermo Fisher Scientific) and MES/SDS running buffer prepared according to the manufacturer's instructions; pre-stained protein markers (SeeBlue Plus 2, Thermo Fisher Scientific) were included. Gels were stained for total protein with SimplyBlue SafeStain (Thermo Fisher Scientific) according to the manufacturer's instructions.

## β-Lactam hydrolysis assay

β-Lactam hydrolysis measurements were carried out using the chromogenic β-lactam nitrocefin (Abcam). Briefly, overnight cultures of strains to be tested were centrifuged, pellets were weighed and resuspended in 150 µL of 100 mM sodium phosphate buffer (pH 7.0) per 1 mg of wet-cell pellet, and cells were lysed by sonication. Lysates were transferred into clear-bottomed 96-well microtiter plates (Corning) at volumes that corresponded to the following weights of bacterial cell pellets: strains harboring pDM1, pDM1-$bla_{L2-1}$, and pDM1-$bla_{OXA-50}$ (0.34 mg of cell pellet); strains harboring pDM1-$bla_{BEL-1}$, pDM1-$bla_{AIM-1}$, and pDM1-$bla_{SMB-1}$ (0.17 mg of cell pellet); strains harboring pDM1-$bla_{POM-1}$ (0.07 mg of cell pellet); strains harboring pDM1-$bla_{BPS-1m}$ (0.07 mg of cell pellet); strains harboring pDM1-$bla_{CARB-2}$ (0.03 mg of cell pellet). In all cases, nitrocefin was added at a final concentration of 400 µM and the final reaction volume was made up to 100 µL using 100 mM sodium phosphate buffer (pH 7.0). Nitrocefin hydrolysis was monitored at 25 °C by recording absorbance at 490 nm at 60 s intervals for 15 minutes using an Infinite M200 Pro microplate reader (Tecan). The amount of nitrocefin hydrolyzed by each lysate in 15 min was calculated using a standard curve generated by acid hydrolysis of nitrocefin standards.

## *Galleria mellonella* survival assay

The wax moth model *G. mellonella* was used for *in vivo* survival assays (*McCarthy et al., 2017*). Individual *G. mellonella* larvae were randomly allocated to experimental groups; no masking was used. Overnight cultures of all the strains to be tested were standardized to $OD_{600}$ 1.0, suspensions were centrifuged, and the pellets were washed three times in PBS and serially diluted. For experiments with *P. aeruginosa* G6R7, 10 µL of the 1:10,000 dilution of each bacterial suspension was injected into the last right abdominal proleg of 40 *G. mellonella* larvae per condition. One hour after infection, larvae were injected with 2.75 µL of piperacillin to a final concentration of 5 µg/mL in the last left abdominal proleg. For experiments with *P. aeruginosa* CDC #773, 10 µL of the 1:1000 dilution of each bacterial suspension was injected into the last right abdominal proleg of 30 *G. mellonella* larvae per condition. Immediately after the injection with the inoculum, the larvae were injected with 4.5 µl of ceftazidime to a final concentration of 6.5 µg/mL in the last left abdominal proleg. All larvae were incubated at 37 °C and their mortality was monitored for 30 hr. Death was recorded when larvae turned black due

to melanization and did not respond to physical stimulation. For each experiment, an additional ten larvae were injected with PBS as negative control, and experiments were discontinued and discounted if mortality was greater than 10% in the PBS control.

## *S. maltophilia - P. aeruginosa* protection assay

The protection assay was based on the approach described in *Bottery et al., 2022*. Briefly, 75 µL of double-strength SCFM medium were transferred into clear-bottomed 96-well microtiter plates (VWR) and inoculated with *S. maltophilia* AMM or its *dsbA dsbL* mutant that had been grown in SCFM medium at 37 °C overnight; *S. maltophilia* strains were inoculated at approximately 5x10⁴, as appropriate. Plates were incubated at 37 °C for 6 hr. Double-strength solutions of ceftazidime at decreasing concentrations were prepared by two-fold serial dilution in sterile ultra-pure $H_2O$ and were added to the wells, as required. *P. aeruginosa* PA14 was immediately added to all the wells at approximately 5x10⁴ CFUs, and the plates were incubated for 20 hr at 37 °C.

To enumerate *P. aeruginosa* in this experiment, the *P. aeruginosa* PA14 *att*Tn*7::accC* strain was used. Following the 20 hr incubation step, serial dilutions of the content of each well were performed in MH broth down to a $10^{-7}$ dilution, plated on MH agar supplemented with gentamicin (*S. maltophilia* AMM strains are sensitive to gentamicin, whereas *P. aeruginosa* PA14 *att*Tn*7::accC* harbors a gentamicin resistance gene on its Tn*7* site) and incubated at 37 °C overnight. CFUs were enumerated the following day. S. maltophilia was enumerated in a separate experiment, whereby *S. maltophilia* AMM *att*Tn*7::accC msfgfp* or its *dsbA dsbL* mutant were used. Following the 20 hr incubation step, serial dilutions of the content of each well were performed in MH broth down to a $10^{-7}$ dilution, plated on MH agar supplemented with gentamicin (*S. maltophilia* AMM strains harbor a gentamicin resistance gene on their Tn*7* site, whereas *P. aeruginosa* PA14 is sensitive to gentamicin) and incubated at 37 °C overnight. CFUs were enumerated the following day.

## Statistical analysis of experimental data

The total number of performed biological experiments and technical repeats is mentioned in the figure legend of each display item. Biological replication refers to completely independent repetition of an experiment using different biological and chemical materials. Technical replication refers to independent data recordings using the same biological sample.

Antibiotic MIC values were determined in biological triplicate, except for MIC values recorded for *dsbA* complementation experiments in our *E. coli* K-12 inducible system that were carried out in duplicate. All ETEST MICs were determined as a single technical replicate, and all BMD MICs were determined in technical triplicate. All recorded MIC values are displayed in the relevant graphs; for MIC assays where three or more biological experiments were performed, the bars indicate the median value, while for assays where two biological experiments were performed, the bars indicate the most conservative of the two values (i.e. for increasing trends, the value representing the smallest increase and for decreasing trends, the value representing the smallest decrease). We note that in line with recommended practice, our MIC results were not averaged. This should be avoided because of the quantized nature of MIC assays, which only inform on bacterial survival for specific antibiotic concentrations and do not provide information for antibiotic concentrations that lie in-between the tested values.

For all other assays, statistical analysis was performed in GraphPad Prism v8.3.1 using either an unpaired T-test with Welch's correction or a Mantel-Cox log-rank test, as appropriate. Statistical significance was defined as $p < 0.05$. Outliers were defined as any technical repeat >2 SD away from the average of the other technical repeats within the same biological experiment. Such data were excluded, and all remaining data were included in the analysis. Detailed information for each figure is also provided below:

*Figure 2B*: unpaired T-test with Welch's correction; n=3; 3.417 degrees of freedom, t-value=0.3927, p=0.7178 (non-significance; for pDM1 strains); 2.933 degrees of freedom, t-value=0.3296, p=0.7639 (non-significance; for pDM1-bla$_{L2-1}$ strains); 2.021 degrees of freedom, t-value=7.549, p=0.0166 (significance; for pDM1-bla$_{BEL-1}$ strains); 2.146 degrees of freedom, t-value=9.153, p=0.0093 (significance; for pDM1-bla$_{CARB-2}$ strains); 2.320 degrees of freedom, t-value=5.668, p=0.0210 (significance; for pDM1-bla$_{AIM-1}$ strains); 3.316 degrees of freedom, t-value=4.353, p=0.0182 (significance; for pDM1-bla$_{OXA-50}$ strains); 3.416 degrees of freedom, t-value=13.⁶⁸, p=0.0004 (significance; for pDM1-bla$_{BPS-1m}$ strains).

*Figure 3C*: Mantel-Cox test; n=40; p=0.3173 (non-significance; *P. aeruginosa* vs *P. aeruginosa dsbA1*), p<0.0001 (significance; *P. aeruginosa* vs *P. aeruginosa* treated with piperacillin), p<0.0001 (significance; *P. aeruginosa* dsbA1 vs *P. aeruginosa* treated with piperacillin), p=0.0147 (significance; *P. aeruginosa* treated with piperacillin vs *P. aeruginosa* dsbA1 treated with piperacillin).

*Figure 3F*: Mantel-Cox test; n=30; p<0.0001 (significance; *P. aeruginosa* vs *P. aeruginosa* dsbA1), p>0.9999 (non-significance; *P. aeruginosa* vs *P. aeruginosa* treated with ceftazidime), p<0.0001 (significance; *P. aeruginosa* dsbA1 vs *P. aeruginosa* treated with ceftazidime), p<0.0001 (significance; *P. aeruginosa* treated with ceftazidime vs *P. aeruginosa* dsbA1 treated with ceftazidime).

*Figure 5—figure supplement 1C*: unpaired T-test with Welch's correction; n=3; 3.417 degrees of freedom, t-value=0.3927, p=0.7178 (non-significance; for pDM1 strains); 2.933 degrees of freedom, t-value=0.3296, p=0.7639 (non-significance; for pDM1-$bla_{L2-1}$ strains); 3.998 degrees of freedom, t-value=4.100, p=0.0149 (significance; for pDM1-$bla_{POM-1}$ strains); 2.345 degrees of freedom, t-value=15.02, p=0.0022 (significance; for pDM1-$bla_{SMB-1}$ strains).

## Bioinformatics

The following bioinformatics analyses were performed in this study. Short scripts and pipelines were written in Perl (version 5.18.2) and executed on macOS Sierra 10.12.5.

### β-Lactamase enzymes

All available protein sequences of β-lactamases were downloaded from http://www.bldb.eu (*Naas et al., 2017*; November 29, 2024). Sequences were clustered using the ucluster software with a 90% identity threshold and the cluster_fast option (USEARCH v.7.0 *Edgar, 2010*); the centroid of each cluster was used as a cluster identifier for every sequence. All sequences were searched for the presence of cysteine residues using a Perl script. Proteins with two or more cysteines after the first 30 amino acids of their primary sequence were considered potential substrates of the DSB system for organisms where oxidative protein folding is carried out by DsbA and provided that translocation of the β-lactamase outside the cytoplasm is performed by the Sec system. The first 30 amino acids of each sequence were excluded to avoid considering cysteines that are part of the signal sequence mediating the translocation of these enzymes outside the cytoplasm. The results of the analysis can be found in *Supplementary file 1*.

### *Stenotrophomonas* MCR-like enzymes

Hidden Markov Models built with validated sequences of MCR-like and EptA-like proteins were used to identify MCR analogues in a total of 106 complete genomes of the *Stenotrophomonas* genus, downloaded from the NCBI repository (March 30, 2023). The analysis was performed with *hmmsearch* (HMMER v.3.1b2; *Finn et al., 2015*) and only hits with evalues <1e-10 were considered. The 146 obtained sequences were aligned using MUSCLE (*Edgar, 2004*), and a phylogenetic tree was built from the alignment using FastTree 2.1.7 with the WAG substitution matrix and the gamma option (*Price et al., 2010*). The assignment of each MCR-like protein sequence to a specific phylogenetic group was carried out based on the best fitting *hmmscan* model. The results of the analysis can be found in *Supplementary file 4*.

## Materials availability

All materials generated in this study are available from the corresponding author upon reasonable request.

## Acknowledgements

We thank L. Dortet for the kind gift of any *P. aeruginosa* and *S. maltophilia* clinical isolates that do not originate from the Centers for Disease Control and Prevention. Publication of this work was supported by the National Institute of Allergy and Infectious Diseases of the National Institutes of Health under Award Number R01AI158753 (to DAIM); the content is solely the responsibility of the authors and does not necessarily represent the official views of the National Institutes of Health. Additionally, this study was funded by the Medical Research Council Career Development Award MR/M009505/1 (to DAIM), a Texas Biologics (TXBio) grant (Award Number TXB-24-02) from The Cockrell School of

Engineering at The University of Texas at Austin (to DAIM), the UT | Portugal Extra Exploratory Project grant 2022.15740.UTA from the Fundação para a Ciência e a Tecnologia, IP (to DAIM), and the Welch Foundation grant F-2250-20250403 (to DAIM), the institutional Biotechnology and Biological Sciences Research Council (BBSRC)-Doctoral Training Program studentship BB/M011178/1 (to NK), the MCIN/ AEI/10.13039/501100011033 Spanish agency through the Ramon y Cajal RYC2019-026551-I and PID2021-123000OB-I00 grants (to PB), the Indiana University Bloomington start-up funds (to CL), and the Cystic Fibrosis Foundation through the Pilot and Feasibility Award 004846I222 (to CL), the Swiss National Science Foundation Ambizione Fellowship PZ00P3_180142 (to DG), as well as the NC3Rs grant NC/V001582/1 (to EM and RRMC), the BBSRC New Investigator Award BB/V007823/1 (to RRMC), the Academy of Medical Sciences / the Wellcome Trust / the Government Department of Business, Energy and Industrial Strategy / the British Heart Foundation / Diabetes UK Springboard Award SBF006\1040 (to RRMC), and the Medical Research Council grant MR/Y001354/1 (to RRMC).

## Additional information

### Funding

| Funder | Grant reference number | Author |
|---|---|---|
| National Institutes of Health | R01AI158753 | Despoina AI Mavridou |
| Medical Research Council | MR/M009505/1 | Despoina AI Mavridou |
| Texas Biologics | TXB-24-02 | Despoina AI Mavridou |
| Portugal Extra Exploratory Project | 2022.15740.UTA | Despoina AI Mavridou |
| Welch Foundation | F-2250-20250403 | Despoina AI Mavridou |
| Biotechnology and Biological Sciences Research Council | BB/M011178/1 | Nikol Kadeřábková |
| MCIN/ AEI/10.13039/501100011033 Spanish agency | RYC2019-026551-I | Patricia Bernal |
| MCIN/ AEI/10.13039/501100011033 Spanish agency | PID2021-123000OB-I00 | Patricia Bernal |
| Cystic Fibrosis Foundation | 004846I222 | Cristina Landeta |
| Swiss National Science Foundation | PZ00P3_180142 | Diego Gonzalez |
| National Centre for the Replacement Refinement and Reduction of Animals in Research | NC/V001582/1 | Ronan R McCarthy |
| Biotechnology and Biological Sciences Research Council | BB/V007823/1 | Ronan R McCarthy |
| Academy of Medical Sciences | SBF006\1040 | Ronan R McCarthy |
| Medical Research Council | MR/Y001354/1 | Ronan R McCarthy |
| Diabetes UK | SBF006\1040 | Ronan R McCarthy |
| British Heart Foundation | SBF006\1040 | Ronan R McCarthy |
| The Government Department of Business, Energy and Industrial Strategy | SBF006\1040 | Ronan R McCarthy |

| Funder | Grant reference number | Author |
|---|---|---|
| Wellcome Trust | SBF006\1040 | Ronan R McCarthy |

The funders had no role in study design, data collection and interpretation, or the decision to submit the work for publication. For the purpose of Open Access, the authors have applied a CC BY public copyright license to any Author Accepted Manuscript version arising from this submission.

## Author contributions

Nikol Kadeřábková, Conceptualization, Data curation, Formal analysis, Investigation, Methodology, Writing – original draft, Writing – review and editing; R Christopher D Furniss, Conceptualization, Supervision, Methodology, Writing – original draft, Writing – review and editing; Evgenia Maslova, Data curation, Formal analysis, Methodology, Writing – review and editing; Kathryn E Potter, Resources; Lara Eisaiankhongi, Data curation, Formal analysis; Patricia Bernal, Resources, Funding acquisition, Methodology, Writing – review and editing; Alain Filloux, Resources, Writing – review and editing; Cristina Landeta, Resources, Funding acquisition, Writing – review and editing; Diego Gonzalez, Data curation, Formal analysis, Funding acquisition, Investigation, Methodology, Writing – review and editing; Ronan R McCarthy, Supervision, Funding acquisition, Methodology, Writing – review and editing; Despoina Al Mavridou, Conceptualization, Supervision, Funding acquisition, Investigation, Methodology, Writing – original draft, Project administration, Writing – review and editing

## Author ORCIDs

Nikol Kadeřábková ⓘ https://orcid.org/0000-0002-1205-1644
Patricia Bernal ⓘ https://orcid.org/0000-0002-6228-0496
Alain Filloux ⓘ https://orcid.org/0000-0003-1307-0289
Cristina Landeta ⓘ https://orcid.org/0000-0003-4065-9646
Despoina Al Mavridou ⓘ https://orcid.org/0000-0002-7449-1151

Reviewer #1 (Public review): https://doi.org/10.7554/eLife.91082.3.sa1
Reviewer #2 (Public review): https://doi.org/10.7554/eLife.91082.3.sa2
Reviewer #3 (Public review): https://doi.org/10.7554/eLife.91082.3.sa3
Author response https://doi.org/10.7554/eLife.91082.3.sa4

# Additional files

## Supplementary files

Supplementary file 1. Analysis of the cysteine content and phylogeny of all identified β-lactamases. 7741 unique β-lactamase protein sequences were clustered with a 90% identity threshold and the centroid of each cluster was used as a phylogenetic cluster identifier for each sequence ('Phylogenetic cluster (90% ID)' column). All sequences were searched for the presence of cysteine residues ('Total number of cysteines' and 'Positions of all cysteines' columns). Proteins with two or more cysteines after the first 30 amino acids of their primary sequence (cells shaded in gray in the 'Number of cysteines after position 30' column) are potential substrates of the DSB system for organisms where oxidative protein olding is carried out by DsbA and provided that translocation of the β-lactamase outside the cytoplasm is performed by the Sec system. The first 30 amino acids of each sequence were excluded to avoid considering cysteines that are part of the signal sequence mediating the translocation of these enzymes outside the cytoplasm. Cells shaded in gray in the 'Reported in pathogens' column mark β-lactamases that are found in pathogens or organisms capable of causing opportunistic infections. The Ambler class of each enzyme is indicated in the 'Ambler class column' and each class (A, B1, B2, B3, C, and D) is highlighted in a different color.

Supplementary file 2. Data used to generate *Figure 1*, *Figure 5—figure supplement 1*, *Figure 4E*, *Figure 4—figure supplement 1B*. (**A**) MIC values (µg/mL) used to generate *Figure 1* are in rows 2–7 (strains serving as negative controls; *E. coli* MC1000 strains harboring pDM1 (vector alone), pDM1-*bla*$_{L2-1}$ or pDM1-*bla*$_{LUT-1}$ (cysteine-containing β-lactamases which lack disulfide bonds)) and rows 9–20. MIC values (µg/mL) used to generate *Figure 5—figure supplement 1* are in rows 22–25. The aminoglycoside antibiotic gentamicin serves as a negative control for all strains. Cells marked with a dash (-) represent strain-antibiotic combinations that were not tested. (**B**) *P. aeruginosa* PA14 colony

forming unit (CFU) counts used to generate *Figure 4E*. (**C**) *P. aeruginosa* PA14, *S. maltophilia* AMM, and *S. maltophilia AMM dsbA dsbL* CFU counts used to generate *Figure 4—figure supplement 1B*. For all tabs, three biological experiments are shown; for (B) and (C), each biological replicate was conducted in technical triplicate and mean CFU values are shown.

Supplementary file 3. Full immunoblots and SDS-PAGE analysis of the immunoblot samples for total protein content. (Pages 1–6). Full immunoblots for *Figure 2A*, *Figure 5—figure supplement 1B*. On the left of each page, the relevant figure panel is shown and the lanes in question are marked with red outline. On the right of each page, the full immunoblot is displayed with the corresponding area also marked with red outline. (**Pages 7–9**) SDS PAGE analysis of the immunoblot samples for total protein content. In each page, the immunoblot in question is indicated (by "*Figure 2A*" or "*Figure 5—figure supplement 1B*") and lanes are marked accordingly to identify the immunoblot lane that they correspond to (see white labels at the bottom of the gel).

Supplementary file 4. Analysis of *Stenotrophomonas spp.* for the presence of MCR proteins. Hidden Markov Models built from validated sequences of MCR-like and EptA-like proteins were used for the identification of MCR-like analogs in a total of 106 complete genomes of the *Stenotrophomonas* genus downloaded from the NCBI repository. (**A**) Most genomes that were investigated ('*Stenotrophomonas maltophilia* genome' column), encoded one or two MCR-like proteins ('Number of MCR analogues column'). (**B**) The 146 MCR-like sequences ('Protein ID column') that were identified (only hits with evalues <1e-10 were considered; 'Evalue' column) belong to the same phylogenetic group as validated MCR-5 or MCR-8 proteins ('Phylogenetic group' column).

Supplementary file 5. Quality control information on 4,5-dibromo-2-(2-chlorobenzyl)pyridazin-3(2 H)-one. $^1$H-NMR and LCMS spectra of 4,5-dibromo-2-(2-chlorobenzyl)pyridazin-3(2 H)-one (compound 36) demonstrating the correctness and purity of the synthesized compound by Bioduro-Sundia.

Supplementary file 6. Supplemary file including supplementary Tables 1 - 5 and supplementary reference citations.

MDAR checklist

### Data availability

All data generated or analyzed during this study are included in the manuscript and supporting files; source data files have been provided.

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

# Appendix 1

## Appendix 1—key resources table

| Reagent type (species) or resource | Designation | Source or reference | Identifiers | Additional information |
|---|---|---|---|---|
| Genetic reagent (*Escherichia coli*) | DH5α | *Hanahan, 1985* | F⁻ *endA1 glnV44 thi-1 recA1 relA1 gyrA96 deoR nupG purB20 φ80dlacZΔM15 Δ(lacZYA-argF)U169 hsdR17(r$_K^-$m$_K^+$) λ⁻* | - |
| Genetic reagent (*E. coli*) | DH5α λ pir | *Martínez-García and de Lorenzo, 2011* | λ pir | - |
| Genetic reagent (*E. coli*) | CC118 λ pir | *Herrero et al., 1990* | *araD Δ(ara, leu) ΔlacZ74 phoA20 galK thi-1 rspE rpoB argE recA1 λ pir* | - |
| Genetic reagent (*E. coli*) | HB101 | *Boyer and Roulland-Dussoix, 1969* | *supE44 hsdS20 recA13 ara-14 proA2 lacY1 galK2 rpsL20 xyl-5 mtl-1* | - |
| Genetic reagent (*E. coli*) | MC1000 | *Casadaban and Cohen, 1980* | *araD139 Δ(ara, leu)7697 ΔlacX74 galU galK strA* | - |
| Genetic reagent (*E. coli*) | MC1000 *dsbA* | *Kadokura et al., 2004* | *dsbA::aphA*, Kan$^R$ | - |
| Genetic reagent (*E. coli*) | MC1000 *dsbA attTn7::Ptac-dsbA* | *Furniss et al., 2022* | *dsbA::aphA attTn7::dsbA*, Kan$^R$ | Can be obtained from the Mavridou lab |
| Strain, strain background (*Pseudomonas aeruginosa*) | PAO1 | *Holloway, 1969* | wild-type prototroph | - |
| Strain, strain background (*P. aeruginosa*) | PA14 | *Rahme et al., 1995* | wild-type prototroph | - |
| Genetic reagent (*P. aeruginosa*) | PA14 *attTn7::accC* | This study | *attTn7::accC*, Gent$^R$ | Can be obtained from the Mavridou lab |
| Strain, strain background (*P. aeruginosa*) | G4R7 | *Dortet et al., 2018* | *bla*$_{AIM-1}$ | Human clinical strain |
| Genetic reagent (*P. aeruginosa*) | G4R7 *dsbA1* | This study | *dsbA1 bla*$_{AIM-1}$ | Can be obtained from the Mavridou lab |
| Strain, strain background (*P. aeruginosa*) | G6R7 | *Dortet et al., 2018* | *bla*$_{AIM-1}$ | Human clinical strain |
| Genetic reagent (*P. aeruginosa*) | G6R7 *dsbA1* | This study | *dsbA1 bla*$_{AIM-1}$ | Can be obtained from the Mavridou lab |
| Strain, strain background (*P. aeruginosa*) | CDC #769 | CDC AR Isolate Bank | *bla*$_{GES-19}$*bla*$_{GES-26}$ | Human clinical strain |
| Genetic reagent (*P. aeruginosa*) | CDC #769 *dsbA1* | This study | *dsbA1 bla*$_{GES-19}$*bla*$_{GES-20}$ | Can be obtained from the Mavridou lab |
| Strain, strain background (*Stenotrophomonas maltophilia*) | AMM | *Emeraud et al., 2019* | *bla*$_{L2-1}$*bla*$_{L1-1}$ | Human clinical strain |
| Genetic reagent (*S. maltophilia*) | AMM *dsbA dsbL* | This study | *dsbA dsbL bla*$_{L2-1}$*bla*$_{L1-1}$ | Can be obtained from the Mavridou lab |
| Genetic reagent (*S. maltophilia*) | AMM *attTn7::accC msfgfp* | This study | *bla*$_{L2-1}$*bla*$_{L1-1}$ *attTn7::accC msfgfp*, Gent$^R$ | Can be obtained from the Mavridou lab |
| Genetic reagent (*S. maltophilia*) | AMM *dsbA dsbL attTn7::accC msfgfp* | This study | *dsbA dsbL bla*$_{L2-1}$*bla*$_{L1-1}$*attTn7::accC msfgfp*, Gent$^R$ | Can be obtained from the Mavridou lab |
| Genetic reagent (*S. maltophilia*) | AMM *dsbA dsbL attTn7::accC msfgfp dsbA1* | This study | *dsbA dsbL bla*$_{L2-1}$*bla*$_{L1-1}$*attTn7::accC msfgfp dsbA1*, Gent$^R$ | Can be obtained from the Mavridou lab |

*Appendix 1 Continued on next page*

*Appendix 1 Continued*

| Reagent type (species) or resource | Designation | Source or reference | Identifiers | Additional information |
|---|---|---|---|---|
| Strain, strain background (*S. maltophilia*) | GUE | *Emeraud et al., 2019* | *bla*$_{L2-1}$*bla*$_{L1-1}$ | Human clinical strain |
| Genetic reagent (*S. maltophilia*) | GUE *dsbA dsbL* | This study | *dsbA dsbL bla*$_{L2-1}$*bla*$_{L1-1}$ | Can be obtained from the Mavridou lab |
| Recombinant DNA reagent | pDM1 (plasmid) | Mavridou lab stock | GenBank MN128719 | pDM1 vector, p15A *ori*, Ptac promoter, MCS, Tet$^R$ |
| Recombinant DNA reagent | pDM1-*bla*$_{L2-1}$ (plasmid) | *Furniss et al., 2022* | - | *bla*$_{L2-1}$ cloned into pDM1, Tet$^R$; can be obtained from the Mavridou lab |
| Recombinant DNA reagent | pDM1-*bla*$_{LUT-1}$ (plasmid) | This study | - | *bla*$_{LUT-1}$ cloned into pDM1, Tet$^R$; can be obtained from the Mavridou lab |
| Recombinant DNA reagent | pDM1-*bla*$_{CARB-2}$ (plasmid) | This study | - | *bla*$_{CARB-2}$ cloned into pDM1, Tet$^R$; can be obtained from the Mavridou lab |
| Recombinant DNA reagent | pDM1-*bla*$_{BPS-1m}$ (plasmid) | This study | - | *bla*$_{BPS-1m}$ cloned into pDM1, Tet$^R$; can be obtained from the Mavridou lab |
| Recombinant DNA reagent | pDM1-*bla*$_{BPS-6}$ (plasmid) | This study | - | *bla*$_{BPS-6}$ cloned into pDM1, Tet$^R$; can be obtained from the Mavridou lab |
| Recombinant DNA reagent | pDM1-*bla*$_{AIM-1}$ (plasmid) | This study | - | *bla*$_{AIM-1}$ cloned into pDM1, Tet$^R$; can be obtained from the Mavridou lab |
| Recombinant DNA reagent | pDM1-*bla*$_{POM-1}$ (plasmid) | This study | - | *bla*$_{POM-1}$ cloned into pDM1, Tet$^R$; can be obtained from the Mavridou lab |
| Recombinant DNA reagent | pDM1-*bla*$_{SMB-1}$ (plasmid) | This study | - | *bla*$_{SMB-1}$ cloned into pDM1, Tet$^R$; can be obtained from the Mavridou lab |
| Recombinant DNA reagent | pDM1-*bla*$_{OXA-50}$ (plasmid) | This study | - | *bla*$_{OXA-50}$ cloned into pDM1, Tet$^R$; can be obtained from the Mavridou lab |
| Recombinant DNA reagent | pDM1-*bla*$_{L2-1}$-StrepII (plasmid) | *Furniss et al., 2022* | - | *bla*$_{L2-1}$ encoding L2-1 with a C-terminal StrepII tag cloned into pDM1, Tet$^R$; can be obtained from the Mavridou lab |
| Recombinant DNA reagent | pDM1-*bla*$_{BEL-1}$-StrepII (plasmid) | This study | - | *bla*$_{BEL-1}$ encoding BEL-1 with a C-terminal StrepII tag cloned into pDM1, Tet$^R$; can be obtained from the Mavridou lab |
| Recombinant DNA reagent | pDM1-*bla*$_{CARB-2}$-StrepII (plasmid) | This study | - | *bla*$_{CARB-2}$ encoding CARB-2 with a C-terminal StrepII tag cloned into pDM1, Tet$^R$; can be obtained from the Mavridou lab |

*Appendix 1 Continued on next page*

*Appendix 1 Continued*

| Reagent type (species) or resource | Designation | Source or reference | Identifiers | Additional information |
|---|---|---|---|---|
| Recombinant DNA reagent | pDM1-*bla*BPS-1m -StrepII (plasmid) | This study | - | *bla*BPS-1m encoding BPS-1m with a C-terminal StrepII tag cloned into pDM1, Tet$^R$; can be obtained from the Mavridou lab |
| Recombinant DNA reagent | pDM1-*bla*AIM-1-StrepII (plasmid) | This study | - | *bla*AIM-1 encoding AIM-1 with a C-terminal StrepII tag cloned into pDM1, Tet$^R$; can be obtained from the Mavridou lab |
| Recombinant DNA reagent | pDM1-*bla*POM-1-StrepII (plasmid) | This study | - | *bla*POM-1 encoding POM-1 with a C-terminal StrepII tag cloned into pDM1, Tet$^R$; can be obtained from the Mavridou lab |
| Recombinant DNA reagent | pDM1-*bla*SMB-1-StrepII (plasmid) | This study | - | *bla*SMB-1 encoding SMB-1 with a C-terminal StrepII tag cloned into pDM1, Tet$^R$; can be obtained from the Mavridou lab |
| Recombinant DNA reagent | pDM1-*bla*OXA-50-StrepII (plasmid) | This study | - | *bla*OXA-50 encoding OXA-50 with a C-terminal StrepII tag cloned into pDM1, Tet$^R$; can be obtained from the Mavridou lab |
| Recombinant DNA reagent | pKNG101 (plasmid) | **Kaniga et al., 1991** | - | Gene replacement suicide vector, *ori*R6K, *ori*TRK2, *sacB*, Str$^R$ |
| Recombinant DNA reagent | pKNG102 (plasmid) | Bernal lab stock | - | Gene replacement suicide vector, *ori*R6K, *ori*TRK2, *sacB*, Tet$^R$; can be obtained from the Mavridou lab |
| Recombinant DNA reagent | pKNG101-*dsbA1* (plasmid) | **Furniss et al., 2022** | - | PCR fragment containing the regions upstream and downstream *P. aeruginosa dsbA1* cloned in pKNG101; when inserted into the chromosome, the strain is a merodiploid for *dsbA1* mutant, Str$^R$; can be obtained from the Mavridou lab |
| Recombinant DNA reagent | pKNG102-*dsbA1*-769 (plasmid) | This study | - | PCR fragment containing the regions upstream and downstream *P. aeruginosa* CDC #769 (**Supplementary file 2**) *dsbA1* cloned in pKNG102; when inserted into the chromosome, the strain is a merodiploid for *dsbA1* mutant, Tet$^R$; can be obtained from the Mavridou lab |
| Recombinant DNA reagent | pKD4 (plasmid) | **Datsenko and Wanner, 2000** | - | Conditional oriRγ *ori*, (template for the *aphA* cassette), Amp$^R$ |

*Appendix 1 Continued on next page*

*Appendix 1 Continued*

| Reagent type (species) or resource | Designation | Source or reference | Identifiers | Additional information |
|---|---|---|---|---|
| Recombinant DNA reagent | pCB112 (plasmid) | *Paradis-Bleau et al., 2014* | - | Inducible *lacZ* expression under the control of the P$_{lac}$ promoter, pBR322 *ori*, Cam$^R$ |
| Recombinant DNA reagent | pKNG101 (plasmid) | *Kaniga et al., 1991* | - | Gene replacement suicide vector, *ori*R6K, *ori*TRK2, *sacB*, (template for the *strAB* cassette), Str$^R$ |
| Recombinant DNA reagent | pKNG101-*dsbA1* (plasmid) | This study | - | PCR fragment containing the regions upstream and downstream *P. aeruginosa dsbA1* cloned in pKNG101; when inserted into the chromosome the strain is a merodiploid for *dsbA1* mutant, Str$^R$; can be obtained from the Mavridou lab |
| Recombinant DNA reagent | pKNG102-*dsbA1*-773 | This study | - | PCR fragment containing the regions upstream and downstream *P. aeruginosa* CDC #773 (*Supplementary file 2*) *dsbA1* cloned in pKNG102; when inserted into the chromosome, the strain is a merodiploid for *dsbA1* mutant, Tet$^R$; can be obtained from the Mavridou lab |
| Recombinant DNA reagent | pKNG101-*dsbA dsbL*-AMM (plasmid) | This study | - | PCR fragment containing the regions upstream and downstream *S. maltophilia* AMM *dsbA* and *dsbL* genes cloned in pKNG101; when inserted into the chromosome, the strain is a merodiploid for *dsbA dsbL* mutant, Str$^R$; can be obtained from the Mavridou lab |
| Recombinant DNA reagent | pKNG101-*dsbA dsbL*-GUE (plasmid) | This study | - | PCR fragment containing the regions upstream and downstream *S. maltophilia* GUE *dsbA* and *dsbL* genes cloned in pKNG101; when inserted into the chromosome, the strain is a merodiploid for *dsbA dsbL* mutant, Str$^R$; can be obtained from the Mavridou lab |
| Recombinant DNA reagent | pRK600 (plasmid) | *Kessler et al., 1992* | - | Helper plasmid, ColE1 *ori*, *mob*RK2, *tra*RK2, Cam$^R$ |
| Recombinant DNA reagent | pTn7-M (plasmid) | *Zobel et al., 2015* | - | Mini-Tn7 delivery transposon vector containing the Tn7 flanking regions and a Gent$^R$ marker, R6K *ori*, Kan$^R$, Gent$^R$ |

*Appendix 1 Continued on next page*

*Appendix 1 Continued*

| Reagent type (species) or resource | Designation | Source or reference | Identifiers | Additional information |
|---|---|---|---|---|
| Recombinant DNA reagent | pBG42 (plasmid) | *Zobel et al., 2015* | - | Mini-Tn*7* delivery transposon vector containing the Tn*7* flanking regions, a Gent^R marker and *msfgfp*, R6K *ori*, Kan^R, Gent^R |
| Recombinant DNA reagent | pBG42-PAO1*dsbA1* (plasmid) | This study | - | *dsbA1* encoding DsbA1 from *P. aeruginosa* PAO1 cloned into pBG42, Kan^R, Gent^R; can be obtained from the Mavridou lab |
| Recombinant DNA reagent | pTNS2 (plasmid) | *Choi et al., 2005* | - | Helper plasmid, R6K *ori*; encodes the TnsABC +D specific transposition pathway, Amp^R |
| Recombinant DNA reagent | pMK-RQ *carb-2* (plasmid) | This study | - | GeneArt cloning vector containing *carb-2*, ColE1 *ori*, (template for *carb-2*), Kan^R; can be obtained from the Mavridou lab |
| Recombinant DNA reagent | pMK-RQ *bps-1m* (plasmid) | This study | - | GeneArt cloning vector containing *bps-1m*, ColE1 *ori*, (template for *bps-1m*), Kan^R; can be obtained from the Mavridou lab |
| Recombinant DNA reagent | pMK-RQ *bps-6* (plasmid) | This study | - | GeneArt cloning vector containing *bps-6*, ColE1 *ori*, (template for *bps-6*), Kan^R; can be obtained from the Mavridou lab |
| Recombinant DNA reagent | pMK-RQ *smb-1* (plasmid) | This study | - | GeneArt cloning vector containing *smb-1*, ColE1 *ori*, (template for *smb-1*), Kan^R; can be obtained from the Mavridou lab |
| Chemical compound, drug | Ampicillin | Melford | A40040-10.0 | - |
| Chemical compound, drug | Imipenem | Cambridge Bioscience | CAY16039-100 mg | - |
| Chemical compound, drug | Kanamycin | Gibco | 11815032 | - |
| Chemical compound, drug | Gentamicin | VWR | A1492.0025 | - |
| Chemical compound, drug | Streptomycin | ACROS Organics | AC612240500 | - |
| Chemical compound, drug | Tetracycline | Duchefa Biochemie | T0150.0025 | - |
| Chemical compound, drug | Colistin sulphate | Sigma | C4461-1G | - |
| Chemical compound, drug | Tazobactam | Sigma | T2820-10MG | - |
| Chemical compound, drug | Isopropyl β-D-1-thiogalactopyranoside (IPTG) | Melford | I56000-25.0 | - |
| Chemical compound, drug | KOD Hotstart DNA Polymerase | Sigma | 71086–3 | - |
| Chemical compound, drug | Nitrocefin | Abcam | ab145625-25mg | - |
| Chemical compound, drug | 4,5-dibromo-2-(2-chlorobenzyl) pyridazin-3(2 H)-one | Bioduro-Sundia | - | Custom synthesis |
| Commercial assay or kit | NEBuilder HiFi DNA Assembly | New England Biolabs | E5520S | - |
| Commercial assay or kit | BugBuster Mastermix | Sigma | 71456–3 | - |

*Appendix 1 Continued on next page*

*Appendix 1 Continued*

| Reagent type (species) or resource | Designation | Source or reference | Identifiers | Additional information |
|---|---|---|---|---|
| Commercial assay or kit | SigmaFast BCIP/NBT tablets | Sigma | B5655-25TAB | - |
| Commercial assay or kit | ETEST - Amoxicillin | Biomerieux | 412242 | - |
| Commercial assay or kit | ETEST - Cefuroxime | Biomerieux | 412304 | - |
| Commercial assay or kit | ETEST - Ceftazidime | Biomerieux | 412292 | - |
| Commercial assay or kit | ETEST - Imipenem | Biomerieux | 412373 | - |
| Commercial assay or kit | ETEST - Aztreonam | Biomerieux | 412258 | - |
| Commercial assay or kit | ETEST - Gentamicin | Biomerieux | 412367 | - |
| Antibody | Strep-Tactin-AP conjugate (mouse monoclonal) | Iba Lifesciences | NC0485490 | (1:3,000) in 3 w/v % BSA/TBS-T |
| Antibody | anti-DnaK 8E2/2 (mouse monoclonal) | Enzo Life Sciences | ADI-SPA-880-D | (1:10,000) in 5% w/v skimmed milk/TBS-T |
| Antibody | anti-mouse IgG-AP conjugate (goat polyclonal) | Sigma | A3688-.25ML | (1:6,000) in 5% w/v skimmed milk/TBS-T |
| Software, algorithm | Prism | GraphPad | - | version 8.0.2 |
| Software, algorithm | USEARCH | *Edgar, 2010* | - | version 7.0 |
| Software, algorithm | MUSCLE | *Edgar, 2004* | - | - |
| Software, algorithm | FastTree | *Price et al., 2010* | - | version 2.1.7 |
| Software, algorithm | HMMER | *Finn et al., 2015* | - | version 3.1b2 |

