## [Editor Report · eLife Assessment]

This **important** study demonstrates that disruption of a common protein-folding system renders drug-resistant clinical bacteria susceptible to antibiotics. The work **convincingly** shows that targeting protein folding, including through therapeutic use of small-molecule inhibitors, can be used to combat multidrug-resistant pathogens by potentiating the efficacy of existing drugs. This study is significant and timely as it informs on a new strategy that is relevant to microbiologists and clinicians interested in combating antimicrobial resistance.

---

## [Referee Report · Reviewer #1 (Public review)]

Summary:

In this work the authors provide evidence that impairment of cell envelope protein homeostasis through blocking the machinery for disulfide bond formation restores efficacy of antibiotics including beta-lactam drugs and colistin against AMR in Gram-negative bacteria.

Strengths:

The authors employ a thorough approach to showcase the restoration of antibiotic sensitivity through inhibition of the DSB machinery, including the evaluation of various antibiotics on both normal and Dsb-deficient pathogenic bacteria (i.e. Pseudomonas and Stenotrophomonas). The authors corroborate these findings by employing Dsb inhibitors in addition to delta dsbA strains. The methodology is appropriate and includes measuring MICs as well as validating their observations in vivo using the Galleria model.

---

## [Referee Report · Reviewer #2 (Public review)]

Summary:

This work by Kadeřábková and Furniss et al. demonstrates the importance of a specific protein folding system to effectively folding β-lactamase proteins, which are responsible for resistance to β-lactam antibiotics, and shows that inhibition of this system sensitize multidrug-resistant pathogens to β-lactam treatment. In addition, the authors extend these observations to a two-species co-culture model where β-lactamases provided by one pathogen can protect another, sensitive pathogen from β-lactam treatment. In this model, disrupting the protein folding system also disrupted protection of the sensitive pathogen from antibiotic killing. Overall, the data presented provide a convincing foundation for subsequent investigations and development of inhibitors for β-lactamases and other resistance determinants. This and similar strategies may have application to polymicrobial contexts when molecular interactions are suspected to confer resistance to natively antibiotic-sensitive pathogens.

Strengths:

The authors use clear and reliable molecular biology strategies to show that β-lactamase proteins from *P. aeruginosa* and Burkholderia species, expressed in *E. coli* in the absence of the dsbA protein folding system, are variably less capable of resisting the effects of different β-lactam antibiotics compared to the dsbA-competent parent strain (Figure 1). The appropriate control is included in the supplemental materials to demonstrate that this effect is specifically dependent on dsbA, since complementing the mutant with an intact dsbA gene restores antibiotic resistance (Figure S1). The authors subsequently show that this lack of activity can be explained by significantly reduced protein levels and loss-of-function protein misfolding in the dsbA mutant background (Figure 2). These data support the importance of this protein folding mechanism in the activity of multiple clinically relevant β-lactamases.

Native bacterial species are used for subsequent experiments, and the authors provide important context for their antibiotic choices and concentrations by referencing the breakpoints that guide clinical practice. In Figure 4, the authors show that loss of the DsbA system in *P. aeruginosa* significantly sensitizes clinical isolates expressing different classes of β-lactamases to clinically relevant antibiotics. The appropriate control showing that the dsbA1 mutation does not result in sensitivity to a non-β-lactam antibiotic is included in Figure S2. The authors further show, using an in vivo model for antibiotic treatment, that treatment of a dsbA1 mutant results in moderate and near-complete survival of the infected organisms. The importance of this system in *S. maltophilia* is then investigated similarly (Figure 5), showing that a dsbA dsbL mutant is also sensitive to β-lactams and colistin, another antibiotic whose resistance mechanism is dependent on the DsbA protein folding system. Importantly, the authors show that a small-molecule inhibitor that disrupts the DsbA system, rather than genetic mutations, is also capable of sensitizing S. maltophilia to these antibiotics. It should be noted that while the sensitization is less pronounced, this molecule has not been optimized for S. maltophilia and would be expected to increase in efficacy following optimization. Together, the data support that interference with the DsbA system in native hosts can sensitize otherwise resistant pathogens to clinically relevant antibiotic therapy.

Finally, the authors investigate the effects of co-culturing *S. maltophilia* and *P. aeruginosa* (Figure 5E). These assays are performed in synthetic cystic fibrosis sputum medium (SCFM), which provides a nutritional context similar to that in CF but without the presence of more complex components such as mucin. The authors show that while *P. aeruginosa* alone is sensitive to the antibiotic, it can survive moderate concentrations in the presence of *S. maltophilia* and even grow in higher concentrations where S. maltophilia appears to overproduce its β-lactamases. However, this protection is lost in S. maltophilia without the DsbA protein folding system, showing that the protective effect depends on functional production of β-lactamase in the presence of viable S. maltophilia. The authors further achieved the difficult task of labeling these multi-drug resistant pathogens with selection markers to determine co-infection CFUs in the supplemental materials. Overall, the data support a protective role for DsbA-dependent β-lactamase under these co-culture conditions.

Weaknesses:

No significant weaknesses are noted beyond the limitations identified and discussed by the authors.

---

## [Referee Report · Reviewer #3 (Public review)]

Summary:

In the face of emerging antibiotic resistance and slow pace of drug discovery, strategies that can enhance the efficacy of existing clinically used antibiotics are highly sought after. In this manuscript, through genetic manipulation of a model bacterium (*Escherichia coli*) and clinically isolated and antibiotic resistant strains of concern (Pseudomonas, Burkholderia, Stenotrophomonas), an additional drug target to combat resistance and potentiate existing drugs is put forward. These observations were validated in both pure cultures, mixed bacterial cultures and in worm models. The drug target investigated in this study appears to be broadly relevant to the challenge posed by lactamases enzyme that render lactam antibiotics ineffective in the clinic. The compounds that target this enzyme are being developed already, some of which were tested in this study displaying promising results and potential for further optimization by medicinal chemists.

Strengths:

The work is well designed and well executed and targets an urgent area of research with the unprecedented increase in antibiotic resistance.

Weaknesses:

The impact of the work can be strengthened by demonstrating increased efficacy of antibiotics in mice models or wound models for Pseudomonas infections. Worm models are relevant, but still distant from investigations in animal models.

---

## [Author Response]

The following is the authors’ response to the current reviews.

**Reviewer #1 (Recommendation For the Authors):**
Thanks to the authors for addressing my suggestions. I think these modifications have improved the clarity of the data and the overall presentation of the manuscript. The methods are now more clearly explained, and the additional details help make the results easier to interpret. Where addressing the comment wasn't feasible, the authors gave reasonable explanations. Overall, the revisions strengthen the paper, and I have no further concerns.

Thank you for your recommendations, which have significantly improved our paper.

**Reviewer #2 (Recommendation For the Authors):**
The additional work conducted by the authors is greatly appreciated. All concerns (and beyond) have been thoroughly addressed by the authors and I am thankful for their consideration and attention to detail. Only one possible issue with the revisions is described below for consideration:Regarding the CFU counts and/or axis labels in Figure S3B, some of the listed "CFU per 1 mL" values (in both the figure itself and File S2B) are extraordinarily high. For example, the greatest CFU for PA14 observed in Figure 4E is ~1x10^9. However, PA14 at 0 ug/mL Ceftazidime reaches nearly 1x10^16 in Figure S3B. From what I can tell, this should be beyond the capacity of bacteria in this space by several orders of magnitude. (E.g., a cubic centimeter [~1 mL] is ~1x10^12 cubic micrometers. At their smallest dimensions and volume, a maximum of ~1x10^13 cells could theoretically fit in this space assuming no liquid and perfect organization.) Similarly, both "AMM" and "AMM (+PA14)" consistently reach CFUs between 1x10^12 and 1x10^14 in this assay. Are the authors confident in the values and/or depiction of CFUs for this figure? It seems like this could be a labeling or dilutioncounting issue.

Thank you for your positive remarks on our revised manuscript and for your constructive comments that have strengthened our work.

We agree with the concern regarding the CFU counts in Figure S3B. The very high values (>10^12^CFU) reflect a technical enumeration artifact that, due to the nature of the assay, cannot be fully avoided. The origin of these inflated counts is described in more detail below:

Following competition assays between *Pseudomonas aeruginosa* and *Stenotrophomonas maltophilia* in liquid culture with antibiotics, we enumerate survivors for each species by colony forming unit (CFU) counts. Because two different bacterial species must be quantified from mixed cultures, we use a gentamicin resistance marker carried by one species at a time.

Each condition is therefore enumerated twice, as we alternate which species harbors the gentamicin cassette.

During coculture in antibiotics and minimal medium, clinical isolates of *P. aeruginosa* and *S. maltophilia*, like those used here, can transiently increase their tolerance to antibiotics, including aminoglycosides. This reduces the effectiveness of gentamicin selection at the plating step necessary for CFU enumeration. For the data presented in Figure S3B, in a subset of highOD₆₀₀ conditions in the competition assay, this tolerance produces artificially inflated CFU values that exceed the biological carrying capacity during the CFU enumeration step.

We evaluated alternative enumeration strategies (e.g., fluorescent protein markers with a nonselective medium), but these proved unsuitable for these strains due to differences in growth rates and media compatibility, introducing other large biases. Given these constraints, selective plating remains the only feasible approach for this work, and the associated artifact cannot be eliminated entirely.

Importantly, transient resistance (tolerance), although common, is not a universal occurrence (e.g., we did not observe it when we performed the experiments shown in Figure 4E). When it does arise, it occurs reproducibly under the same experimental high-OD_600_ conditions and does not obscure any of the relative comparisons that underpin our conclusions.

For transparency, we have retained the measured values in Figure S3B and we note in the legend that counts above ~10^12^ CFU represent a technical overestimation due to transient gentamicin tolerance. Counts below 10^12^ CFU are accurately enumerated.

**Reviewer #3 (Recommendation For the Authors):**
All concerns have been satisfied and the manuscript is ready for publishing.

Thank you for your recommendations, which have significantly improved our paper.

The following is the authors’ response to the original reviews.

**Reviewer #1 (Public review):**
The study would benefit from presenting raw data in some cases, such as MIC values and SDS-PAGE gels, by clarifying the number of independent experiments used, as well as further clarification on statistical significance for some of the data.

All original data used to generate Fig. 1, Fig. 4E, Fig. S3 and Fig. S4A are presented in File S2. Tab (A) is dedicated to data used for Fig. 1 and Fig. S4A, while tabs (B) and (C) show the data used for Fig. 4E and S3, respectively. This information is indicated in the legends of the relevant figures.

All experiments in this study were performed in three independent (biological) experiments (with the exception of the complementation data shown in Fig. S1 and Fig. S5, which were performed in two independent (biological) experiments). The number of biological and technical replicates for each experiment is stated in the figure legends, as well as in the “Statistical analysis of experimental data” part of the “Materials and Methods” section of the paper. Specifically, for antibiotic MIC assays we have not performed statistical analyses as per recommended practice. The reason for this is stated in the following section from the “Statistical analysis of experimental data” part of the “Materials and Methods” section of the paper (lines 699-711 of the revised manuscript):

“Antibiotic MIC values were determined in biological triplicate, except for MIC values recorded for dsbA complementation experiments in our *E. coli* K-12 inducible system that were carried out in duplicate. All ETEST MICs were determined as a single technical replicate, and all BMD MICs were determined in technical triplicate. All recorded MIC values are displayed in the relevant graphs; for MIC assays where three or more biological experiments were performed, the bars indicate the median value, while for assays where two biological experiments were performed the bars indicate the most conservative of the two values (i.e., for increasing trends, the value representing the smallest increase and for decreasing trends, the value representing the smallest decrease). We note that in line with recommended practice, our MIC results were not averaged. This should be avoided because of the quantized nature of MIC assays, which only inform on bacterial survival for specific antibiotic concentrations and do not provide information for antibiotic concentrations that lie in-between the tested values.”

**Reviewer #2 (Public review):**
While Figure 5E demonstrates a protective effect of DsbA-dependent β-lactamase, the omission of CFU data for *S. maltophilia* makes it difficult to assess the applicability of the polymicrobial strategy. Since S. maltophilia is pre-cultured prior to the addition of *P. aeruginosa* and antibiotics, it is unclear whether the protective effect is dependent on high *S. maltophilia* CFU. It is also unclear what the fate of the S. maltophilia dsbA dsbL mutant is under these conditions. If DsbA-deficient S. maltophilia CFU is not impacted, then this treatment will result in the eradication of only one of the pathogens of interest. If the mutant is lost during treatment, then it is not clear whether the loss of protection is due specifically to the production of non-functional β-lactamase or simply the absence of S. maltophilia.

We have simultaneously tracked the abundance of *P. aeruginosa* and *S. maltophilia* strains in our cross-protection experiment for select antibiotic concentrations. To be able to perform this experiment, we had to label two extremely-drug-resistant strains of S. maltophilia with an antibiotic resistance marker that allowed us to quantify them in mixtures with *P. aeruginosa*. Our results can be found in Fig. S3 of our revised manuscript and, in a nutshell, show that ceftazidime treatment leads to eradication of both *P. aeruginosa* and *S. maltophilia* when disulfide bond formation is impaired in *S. maltophilia*.

The following text was added to address the questions of the reviewer:

“Due to the naturally different growth rates of these two species (*S. maltophilia* grows much slower than *P. aeruginosa*) especially in laboratory conditions, the protocol we followed [1] requires *S. maltophilia* to be grown for 6 hours prior to co-culturing it with *P. aeruginosa*. To ensure that at this point in the experiment our two *S. maltophilia* strains, with and without dsbA, had grown comparatively to each other, we determined their cell densities (Fig. S3A). We found that S. maltophilia AMM dsbA dsbL had grown at a similar level as the wild-type strain, and both were at a higher cell density [~10^7^ colony forming units (CFUs)] compared to the *P. aeruginosa* PA14 inoculum (5 x 10^4^ CFUs)” (lines 353-361 of the revised manuscript).

“To ensure that ceftazidime treatment leads to eradication of both *P. aeruginosa* and *S. maltophilia* when disulfide bond formation is impaired in *S. maltophilia*, we monitored the abundance of both strains in each synthetic community for select antibiotic concentrations (Fig. S3B). In this experiment we largely observed the same trends as in Fig. 4E. At low antibiotic concentrations, for example 4 μg/mL of ceftazidime, S. maltophilia AMM is fully resistant and thrives, thus outcompeting *P. aeruginosa* PA14 (dark pink and dark blue bars in Fig. S3B). The same can also be seen in Fig. 4E, whereby decreased *P. aeruginosa* PA14 CFUs are recorded. By contrast *S. maltophilia* AMM dsbA dsbL already displays decreased growth at 4 μg/mL of ceftazidime because of its non-functional L1-1 enzyme, allowing comparatively higher growth of *P. aeruginosa* (light pink and light blue bars in Fig. S3B). Despite the competition between the two strains, *P. aeruginosa* PA14 benefits from *S. maltophilia* AMM’s high hydrolytic activity against ceftazidime, which allows it to survive and grow in high antibiotic concentrations even though it is not resistant (see 128 μg/mL; dark pink and dark blue bars in Fig. S3B). In stark opposition, without its disulfide bond in *S. maltophilia* AMM dsbA dsbL, L1-1 cannot confer resistance to ceftazidime, resulting in killing of S. maltophilia AMM dsbA dsbL and, consequently, also of *P. aeruginosa* PA14 (see 128 μg/mL; light pink and light blue bars in Fig. S3B).

The data presented here show that, at least under laboratory conditions, targeting protein homeostasis pathways in specific recalcitrant pathogens has the potential to not only alter their own antibiotic resistance profiles (Fig. 3 and 4A-D), but also to influence the antibiotic susceptibility profiles of other bacteria that co-occur in the same conditions (Fig. 5). Admittedly, the conditions in a living host are too complex to draw direct conclusions from this experiment. That said, our results show promise for infections, where pathogen interactions affect treatment outcomes, and whereby their inhibition might facilitate treatment” (lines 381406 of the revised manuscript).

The alleged clinical relevance and immediate, theoretical application of this approach should be properly contextualized. At multiple junctures, the authors state or suggest that interactions between *S. maltophilia* and *P. aeruginosa* are known to occur in disease or have known clinical relevance related to treatment failure and disease states. For instance, the citations provided for *S. maltophilia* protection of *P. aeruginosa* in the CF lung environment both describe simplified laboratory experiments rather than clinical or in vivo observations. Similarly, the citations provided for both the role of *S. maltophilia* in treatment failure and CF disease severity do not support either claim. The role of *S. maltophilia* in CF is currently unsettled, with more recent work reporting conflicting results that support *S. maltophilia* as a marker, rather than cause, of severe disease. These citations also do not support the suggestion that *S. maltophilia* specifically contributes to treatment failure. While it is reasonable to pursue these ideas as a hypothesis or potential concern, there is no evidence provided that these specific interactions occur in vivo or that they have clinical relevance.

Thank you for your comment. You are entirely correct. We have amended the test throughout our revised manuscript to avoid overstating the role of *S. maltophilia* in CF infections and to reference additional relevant works in the literature. Please find below representative examples of such passages:

“On the other hand, CF microbiomes are increasingly found to encompass *S. maltophilia* [2-4], a globally distributed opportunistic pathogen that causes serious nosocomial respiratory and bloodstream infections [5-7]. *S. maltophilia* is one of the most prevalent emerging pathogens [6] and it is intrinsically resistant to almost all antibiotics, including β-lactams like penicillins, cephalosporins and carbapenems, as well as macrolides, fluoroquinolones, aminoglycosides, chloramphenicol, tetracyclines and colistin. As a result, the standard treatment option for lung infections, i.e., broad-spectrum β-lactam antibiotic therapy, is rarely successful in countering S. maltophilia [7,8], creating a definitive need for approaches that will be effective in eliminating both pathogens” (lines 33-41 of the revised manuscript).

“Of the organisms studied in this work, *S. maltophilia* deserves further discussion because of its unique intrinsic resistance profile. The prognosis of CF patients with *S. maltophilia* lung carriage is still debated [4,9-16], largely because studies with extensive and well-controlled patient cohorts are lacking. This notwithstanding, the therapeutic options against this pathogen are currently limited to one non-β-lactam antibiotic-adjuvant combination, , which is not always effective, trimethoprim-sulfamethoxazole [17-20], and a few last-line β-lactam drugs, like the fifth-generation cephalosporin cefiderocol and the combination aztreonam-avibactam. Resistance to commonly used antibiotics causes many problems during treatment and, as a result, infections that harbor *S. maltophilia* have high case fatality rates [7]. This is not limited to CF patients, as S. maltophilia is a major cause of death in children with bacteremia [5]” (lines 440-450 of the revised manuscript).

**Reviewer #3 (Public review):**
The impact of the work can be strengthened by demonstrating increased efficacy of antibiotics in mice models or wound models for Pseudomonas infections. Worm models are relevant, but still distant from investigations in animal models.

Thank you for this comment. We appreciate the sentiment, and we would have liked to be able to perform experiments in a murine model of infection. There are several reasons that made this not possible, and as a result we used G. mellonella as an informative preliminary in vivo infection model. The DSB proteins have been shown to play a central role in bacterial virulence. Because of this our *P. aeruginosa* and *S. maltophilia* mutant strains are not efficient in establishing an infection, even in a wound model. This could be overcome had we been able to use the chemical inhibitor of the DSB system in vivo, however this also is not possible This is due to the fact that the chemical compound that we use to inhibit the function of DsbA acts on DsbB. Inhibition of DsbB blocks the re-oxidation of DsbA and leads to its accumulation in its inactive reduced form. However, the action of the inhibitor can be bypassed through reoxidation and re-activation of DsbA by small-molecule oxidants such as L-cystine, which are abundant in rich growth media or animal tissues. This makes the inhibitor only suitable for in vitro assays that can be performed in minimal media, where the presence of small-molecule oxidants can be strictly avoided, but entirely unsuitable for an insect or a vertebrate animal model.

**Reviewer #1 (Recommendation For the Authors):**
(1) The analysis of the role of DsbA in the assembly of cysteine-containing β-lactamases is a significant finding. However, in addition to showing the MIC fold difference, I think, it would be important to show the raw data for the actual MIC values obtained for each β-lactamase enzyme/antibiotic combination and in both strains (+ and - dsbA).Also, can the authors clarify whether these experiments were conducted on 3 independent samples (there seems to be some contradicting information in the paper and the supplementary figures). If possible, I would also recommend showing in the figure whether the MIC differences observed were statistically significant.

All original data used to generate Fig. 1, Fig. 4E, Fig. S3 and Fig. S4A are presented in File S2. Tab (A) is dedicated to data used for Fig. 1 and Fig. S4A, while tabs (B) and (C) show the data used for Fig. 4E and S3, respectively. This information is indicated in the legends of the relevant figures.

All experiments in this study were performed in three independent (biological) experiments (with the exception of the complementation data shown in Fig. S1 and Fig. S5, which were performed in two independent (biological) experiments). The number of biological and technical replicates for each experiment is stated in the figure legends, as well as in the “Statistical analysis of experimental data” part of the “Materials and Methods” section of the paper. Specifically, for antibiotic MIC assays we have not performed statistical analyses as per recommended practice. The reason for this is stated in the following section from the “Statistical analysis of experimental data” part of the “Materials and Methods” section of the paper (lines 699-711 of the revised manuscript):

“Antibiotic MIC values were determined in biological triplicate, except for MIC values recorded for dsbA complementation experiments in our *E. coli* K-12 inducible system that were carried out in duplicate. All ETEST MICs were determined as a single technical replicate, and all BMD MICs were determined in technical triplicate. All recorded MIC values are displayed in the relevant graphs; for MIC assays where three or more biological experiments were performed, the bars indicate the median value, while for assays where two biological experiments were performed the bars indicate the most conservative of the two values (i.e., for increasing trends, the value representing the smallest increase and for decreasing trends, the value representing the smallest decrease). We note that in line with recommended practice, our MIC results were not averaged. This should be avoided because of the quantized nature of MIC assays, which only inform on bacterial survival for specific antibiotic concentrations and do not provide information for antibiotic concentrations that lie in-between the tested values.”

(2) For Figure 2A, can the authors provide the full Westerns and ideally the SDS-PAGE gel corresponding to the Westerns where the Β-lactamases and the control DNA-K were detected.

Thank you for this comment. Full immunoblots and SDS PAGE analysis of the immunoblot samples for total protein content are shown in File S3 of our revised manuscript.

(3) For the enzymatic assays, was the concentration of enzyme used "normalised " based on the amount detected in the westerns where possible or was only the total amount of protein considered. When similar amounts of enzyme were added, was the activity still compromised?

The β-lactam hydrolysis assay was normalized based on the weight of the cell pellets (wet cell pellet mass) of the tested strains. This means, that for each enzyme expressed in cells with and without DsbA, strains were normalized to the same weight to volume ratio, and thus strains expressing the same enzyme were only compared to each other.

Because enzyme degradation in the absence of DsbA is a key factor underlying the effects we describe for most of the tested β-lactamases (see Fig. 2A and S4A; no protein band is detected for 5 of the 7 enzymes in the dsbA mutant), it was not possible to normalize our samples based on enzyme levels detected by immunoblot. Normalization based on enzyme amounts would be feasible had we purified each β-lactamase after expression in the two different strain backgrounds (+/- dsbA) assuming sufficient protein amounts could be isolated from the dsbA mutant strain. Nonetheless, we feel that such a comparison would be misleading, since enzyme degradation likely plays the biggest role in the lack of activity observed for most of these enzymes in the absence of DsbA.

(4) Not sure whether Fig 3 is very informative. Perhaps it could be redesigned to better encapsulate the findings in this manuscript (combine figurer 3 and 6 into one). I would also include the chemical structure of the inhibitors used and perhaps include how they block the system by binding to DsbB.

Thank you for this comment. Fig. 3 was combined with Fig. 6 of the submitted manuscript. The new model figure is Fig. 5 in our revised manuscript.

The inhibitor compound used in our study has been extensively characterized in a previous publication [21]. Considering that this inhibitor is not the main focus of our paper, we have avoided showing its chemical structure in any of the main display items. That said, its structure can be found in File S5 of our revised manuscript, which contains the quality control information on this compound. As suggested, we included the following sentence to describe the mode of action of this inhibitor: “Compound 36 was previously shown to inhibit disulfide bond formation in *P. aeruginosa* via covalently binding onto one of the four essential cysteine residues of DsbB in the DsbA-DsbB complex [21]” (lines 309-311 of the revised manuscript).(5) Figure 4: Similar to my comment above showing in the figure whether the differences observed in Figure 4, particularly A-C, are statistically significant (i.e. galleria survival difference in the presence and absence of dsbA) would be beneficial.

As mentioned in our answer to comment 1 above, we have not performed statistical analyses for antibiotic MIC assays because, in line with recommended practice, our MIC results were not averaged (Fig. 3A,B,D,E of our revised manuscript). This should be avoided because of the quantized nature of MIC assays, which only inform on bacterial survival for specific antibiotic concentrations and do not provide information for antibiotic concentrations that lie in-between the tested values. Statistical analysis of G. mellonella survival data (Fig. 3C,F of our revised manuscript) was performed and is described fully in the legend of Fig. 3, as well as in the “Statistical analysis of experimental data” part of the “Materials and Methods” section of the paper (lines 729-738 of the revised manuscript). Finally, the statistical analyses for the most important comparisons in panels (C) and (F) of Fig. 3 are also marked directly on the figure.

(6) Were the authors able to test the redox state of DsbA upon addition of the DsbB inhibitor to further demonstrate that the effects observed were indeed due to the obstruction of the Dsb machinery and not due to off target effects.

Thank you for the opportunity to clarify this. In previous work from our lab, we have used a DSB system inhibitor termed “compound 12” in [22] with activity against DsbB proteins from Enterobacteria. In our previous study [23] we, indeed, tested the redox state of DsbA in the presence of this inhibitor compound. We could not perform the same experiment here with “compound 36” from [21], because we do not have an antibody against the DsbA protein of *S. maltophilia*. That said, we have carried out experiments that confirm that our results are due to specific inhibition of the DSB system and not because of off-target effects. In particular, we show that the gentamicin MIC values of S. maltophilia AMM remain unchanged in the presence of the inhibitor and treatment of S. maltophilia AMM dsbA dsbL with the compound does not affects its colistin MIC value (Fig. S2E and lines 317-320 of the revised manuscript).

(7) Given the remarkable effects shown by the DsbB inhibitor, did the authors use this compound to assess whether inhibition of the Dsb system with small molecules would block cross-resistance in *S. maltophilia* - *P. aeruginosa* mixed communities (Fig 5D).

Unfortunately, this was not possible. The decrease in the ceftazidime MIC value of *S. maltophilia* AMM in the presence of the DSB inhibitor compound is more modest than the effects we observed when the dsbA dsbL mutant is used (compare Fig. 4D (left) with Fig.4A of the revised manuscript). This means that in the presence of the DSB inhibitor there are still sufficient amounts of functional β-lactamase present and we expect that they would contribute to cross-protection of *P. aeruginosa*. While the use of the DSB inhibitor does have a drastic impact on the colistin resistance profile of *S. maltophilia* AMM (Fig. 4D of the revised manuscript), unlike β-lactamases, which act as common goods, MCR enzymes act solely on the lipopolysaccharide of their producer and do not contribute to bacterial interactions, precluding the use of colistin for a cross-protection experiment.

**Reviewer #2 (Recommendation For the Authors):**
(1) The acronym used for synthetic cystic fibrosis sputum medium (lines 523, 531, 535, 601, and 603) is defined in the manuscript as 'SCF', but the common formulation is 'SCFM', including in the provided citation. Suggest changing to SCFM for consistency.

Thank you for this comment. This has been amended throughout our revised manuscript.

(2) In Figure 1, while the legend states that "No changes in MIC values are observed for strains harboring the empty vector control (pDM1)[...]" (lines 729-30), the median of ceftazidime in the pDM1 control appears to indicate a 2-fold decrease in MIC. This would not seem to significantly impact the other results since the MIC decreases observed for other conditions are all 3-fold or greater, but this should be addressed and/or explained in the text.

You are correct. Thank you for the opportunity to clarify this. Generally, since MIC assays have a degree of variability, we have only followed decreases in MIC values that are greater than 2fold. Generally, for most of our controls, the recorded MIC fold changes are below 2-fold. The only exception to this is the ceftazidime MIC drop of the empty-vector control, showing a 2fold change, which we do not consider significant.

To ensure that this is clear in our text and figure legends the following changes were made:

The clause “only differences larger than 2-fold were considered” was added to the text (lines 110-111 of the revised manuscript).

We amended the legend of Fig. 1 accordingly: “No changes in MIC values are observed for the aminoglycoside antibiotic gentamicin (white bars) confirming that absence of DsbA does not compromise the general ability of this strain to resist antibiotic stress. Minor changes in MIC values (≤ 2-fold) are observed for strains harboring the empty vector control (pDM1) or those expressing the class A β-lactamases L2-1 and LUT-1, which contain two or more cysteines (Table S1), but no disulfide bonds (top row)”.

(3) Similarly, in Fig S1E, there appears to be only partial complementation for BPS-1m. Do the authors hypothesize that this observation is related to a folding defect, rather than degradation of protein, as described for BPS-1m for Figure 2?

Thank you for the opportunity to clarify this. You are correct that we only achieve partial complementation for the *E. coli* strain expressing the BPS-1m enzyme from the Burkholderia complex. Despite the fact that the gene for this enzyme was codon optimized, we observed that its expression in *E. coli* is sub-optimal and incurs fitness effects. In fact, to record the data presented in our manuscript the *E. coli* strains had to be transformed anew every time. Considering that the related enzyme BPS-6 does not present any of these challenges, we attribute the partial complementation to technical difficulties with the expression of the bps-1m gene in *E. coli*.

We clarified this by adding the following clause to our manuscript: “we only achieve partial complementation for the dsbA mutant expressing BPS-1m, which we attribute to the fact that expression of this enzyme in *E. coli* is sub-optimal” (lines 132-134 of the revised manuscript).

(4) Lines 204-206: "[...]we deleted the principal dsbA gene, dsbA1 (pathogenic bacteria often encode multiple DsbA analogues [24,25]), in several multidrug-resistant (MDR) *P. aeruginosa* clinical strains (Table S2)". That multiple DsbA analogues are often encoded is good information to provide, but it was unclear from quickly looking at the citations whether Pa is counted among these. Is it expected that all oxidative protein folding in Pa functions through DsbA1? Conveying this information, if possible, may make the impact of the results in this model clearer.

Thank you for this comment. To address it we added the following text to our manuscript:

“To determine whether the effects on β-lactam MICs observed in our inducible system (Fig. 1 and [23]) can be reproduced in the presence of other resistance determinants in a natural context with endogenous enzyme expression levels, we deleted the principal dsbA gene, dsbA1, in several multidrug-resistant (MDR) *P. aeruginosa* clinical strains (Table S2). Pathogenic bacteria often encode multiple DsbA analogues [24,25] and *P. aeruginosa* is no exception. It encodes two DsbAs, but DsbA1 has been found to catalyze the vast majority of the oxidative protein folding reactions taking place in its cell envelope [26]” (lines 172-178 of the revised manuscript).

(5) Regarding the clinical Pa isolates G4R7 and G6R7, have the authors performed any phenotypic testing on these strains to identify differences that might explain the substantial difference in piperacillin MIC? I.e., can these isolates be distinguished by growth rate, genetic markers or expression levels, early or late infection, mucoidy, etc. This is not essential for the current work, but could weigh on the efficacy of this treatment strategy for AIM1expressing clinical isolates. (E.g., the G4R7 dsbA1 strain exhibits a piperacillin MIC still ~2fold higher than WT G6R7).

Thank you for the opportunity to clarify this. For clinical strains used in our study, we have evaluated their antibiotic resistance profiles, but we have not performed any additional phenotypic characterization. There are many reasons that contribute to differences in antibiotic resistance, starting simply from β-lactamase expression levels and extending to organismal effects, like the ones mentioned by the reviewer. Such characterization would fall outside the scope of our paper, especially since we sensitize our tested *P. aeruginosa* clinical isolates for the majority of the β-lactams antibiotics tested.

We acknowledged this by adding the following sentence to our revised manuscript:

“Despite the fact that *P. aeruginosa* G4R7 dsbA1 was not sensitized for piperacillintazobactam, possibly due to the high level of piperacillin-tazobactam resistance of the parent clinical strain, our results across these two isolates show promise for DsbA as a target against β-lactam resistance in *P. aeruginosa*” (lines 191-194 of the revised manuscript).

(6) Lines 180-2: "This shows that without their disulfide bonds, these proteins are unstable and are ultimately degraded by other cell envelope proteostasis components [33]". While it is clear that protein is significantly lost in all cases except for BPS-1m in 2A, the dsbA pDM1bla constructs in 2B appear to all retain non-trivial (>10-fold) nitrocefin hydrolysis activity compared to the dsbA pDM1 control. This does not impact the other results in 2B, but it would seem that a loss-of-function folding defect, as described subsequently for BPS-1m, is also part of the explanation for the observed MIC decreases, and this was not necessarily clear from the quoted passage. This could simply be clarified in the final sentence - that both mechanisms are potentially in play - if the authors agree with that interpretation.

You are correct, thank you for your comment. We amended the text in our revised manuscript as follows:

The data presented so far (Fig. 1 and 2) demonstrate that disulfide bond formation is essential for the biogenesis (stability and/or protein folding) and, in turn, activity of an expanded set of clinically important β-lactamases, including enzymes that currently lack inhibitor options” (lines 158-161 of the revised manuscript).

(7) While it is clear from Figure S2 that the various dsb mutants do not have a general growth defect or collateral sensitivity to another antibiotic, it does not appear that there is an analogous control for the DSB inhibitor demonstrating no growth/toxic effects at the concentration used. This could be provided similarly to Figure S2, using gentamicin as a control antibiotic.

We have carried out experiments that confirm that our results are due to specific inhibition of the DSB system and not because of off-target effects. In particular, we show that the gentamicin MIC values of *S. maltophilia* AMM remain unchanged in the presence of the inhibitor and treatment of S. maltophilia AMM dsbA dsbL with the compound does not affect its colistin MIC value (Fig. S2E and lines 317-320 of the revised manuscript).

(8) Complementation is appropriately provided for experiments with *E. coli*, but are not provided for *P. aeruginosa* or *S. maltophilia*. It should be straightforward to complement in Pa, but is also probably less critical considering the evidence from *E. coli*. However, since the Sm mutant is a gene cluster with two genes, it would seem more imperative to complement this strain. This reviewer is not familiar enough with Sm to know if complementation is routine or feasible with this organism; if not, the controls for the DSB inhibitor should at least be provided.

As mentioned in our response to comment 7 above, we have carried out experiments that confirm that our DSB inhibitor results are due to specific inhibition of the DSB system and not because of off-target effects.

Moreover, in response to this comment, we have further demonstrated that our results are due to the specific interaction of DsbA with β-lactamase enzymes by complementing dsbA deletions in representative clinical strains of multidrug-resistant *Pseudomonas aeruginosa* and extremely-drug-resistant Stenotrophomonas maltophilia. We would like to note here that gene complementation in clinical isolates remains very rare in the literature due to their high levels of resistance and limited genetic tractability. Most of the few complementation examples reported for these two organisms are limited to strains that, although pathogenic, are commonly used in the lab, or to complementation efforts in non-clinical strain systems (for example use of *P. aeruginosa* PA14 for complementation, instead of the focal clinical isolate).

We tested three different complementation strategies, two of which ended up being unsuccessful. After approximately 9 months of work, we succeeded in complementing a representative clinical strain for each organism (*P. aeruginosa* CDC #769 dsbA1 and *S. maltophilia* AMM dsbA dsbL) by inserting the dsbA1 gene from *P. aeruginosa* PAO1 into the Tn7 site on the chromosome. Both clinical strains show full complementation for every antibiotic tested; our complementation results can be found in Fig. S2B,D of the revised manuscript.

The following text was added for *P. aeruginosa* clinical isolates:

We have demonstrated the specific interaction of DsbA with the tested β-lactamase enzymes in our *E. coli* K-12 inducible system using gentamicin controls (Fig. 1 and File S2A) and gene complementation (Fig. S1). To confirm the specificity of this interaction in *P. aeruginosa*, we performed representative control experiments in one of our clinical strains, *P. aeruginosa* CDC #769. We first tested the general ability of *P. aeruginosa* CDC #769 dsbA1 to resist antibiotic stress by recording MIC values against gentamicin, and found it unchanged compared to its parent (Fig. S2A). Gene complementation in clinical isolates is especially challenging and rarely attempted due to the high levels of resistance and lack of genetic tractability in these strains. Despite these challenges, to further ensure the specificity of the interaction of DsbA with tested β-lactamases in *P. aeruginosa*, we have complemented dsbA1 from *P. aeruginosa* PAO1 into *P. aeruginosa* CDC #769 dsbA1. We found that complementation of dsbA1 restores MICs to wild-type values for both tested β-lactam compounds (Fig. S2B) further demonstrating that our results in *P. aeruginosa* clinical strains are not confounded by off-target effects” (lines 226-239 of the revised manuscript).

The following text was added for *S. maltophilia* clinical isolates:

“Since the dsbA and dsbL are organized in a gene cluster in *S. maltophilia*, we wanted to ensure that our results reported above were exclusively due to disruption of disulfide bond formation in this organism. First, we recorded gentamicin MIC values for S. maltophilia AMM dsbA dsbL and found them to be unchanged compared to the gentamicin MICs of the parent strain (Fig. S2C). This confirms that disruption of disulfide bond formation does not compromise the general ability of this organism to resist antibiotic stress. Next, we complemented S. maltophilia AMM dsbA dsbL. The specific oxidative roles and exact regulation of DsbA and DsbL in S. maltophilia remain unknown. For this reason and considering that genetic manipulation of extremely-drug-resistant organisms is challenging, we used our genetic construct optimized for complementing *P. aeruginosa* CDC #769 dsbA1 with dsbA1 from *P. aeruginosa* PAO1 (Fig. S2B) to also complement *S. maltophilia* AMM dsbA dsbL. We based this approach on the fact that DsbA proteins from one species have been commonly shown to be functional in other species [27-30]. Indeed, we found that complementation of S. maltophilia AMM dsbA dsbL with *P. aeruginosa* PAO1 dsbA1 restores MICs to wild-type values for both ceftazidime and colistin (Fig. S2D), conclusively demonstrating that our results in *S. maltophilia* are not confounded by off-target effects” (lines 282-297 of the revised manuscript).

(9) In Figure 5E, the growth inhibition and loss of Pa CFU in 4 ug/mL ceftazidime for the Sm co-culture condition, which is subsequently lost in the Sm dsbA dsbL co-culture, does not appear to be discussed. As Pa is shown to grow fine in monoculture at this concentration, this result should be discussed in relation to the co-culture dynamics. Is it expected or observed that WT Sm is out-competing Pa under this condition and growing to a high CFU/mL? This would seem to have parallels to citation 49.

As requested by this reviewer (see comment 10 below), we simultaneously tracked the abundance of *P. aeruginosa* and *S. maltophilia* strains in our cross-protection experiment. During this process we probed the abundances of the two organisms at 4 µg/mL of ceftazidime. Our results can be seen in Fig. S3B of the revised manuscript. The reviewer is correct and these effects are due to competition between *P. aeruginosa* and *S. maltophilia* with the latter being able to reach very high CFUs in this antibiotic concentration.

The following text on co-culture dynamics was added to our revised manuscript:

At low antibiotic concentrations, for example 4 μg/mL of ceftazidime, *S. maltophilia* AMM is fully resistant and thrives, thus outcompeting *P. aeruginosa* PA14 (dark pink and dark blue bars in Fig. S3B). The same can also be seen in Fig. 4E, whereby decreased *P. aeruginosa* PA14 CFUs are recorded. By contrast *S. maltophilia* AMM dsbA dsbL already displays decreased growth at 4 μg/mL of ceftazidime because of its non-functional L1-1 enzyme, allowing comparatively higher growth of *P. aeruginosa* (light pink and light blue bars in Fig. S3B)” (lines 384-390 of the revised manuscript).

(10) The data presented in Figure 5E would be augmented by the inclusion of, for at least a few representative cases, the Sm CFUs relative to the Pa CFUs. In describing the protective effects of Sm on Pa for imipenem treatment, the authors of citation 12 note that the effect was dependent on Sm cell density. This raises the immediate question of whether the protection observed in this work is similarly dependent on cell density of Sm. It is unclear if the authors expect Sm to persist under these conditions, and it seems Sm CFU should be expected to be relatively high considering it is pre-incubated for 6 hours prior to the assay. What is the physiological state of these cells, and how are they affected by ceftazidime? While many other variables are likely relevant to the translation of this protection, the relative abundance and localization of Sm and Pa commonly observed in CF patients, as well as the effective concentration of antibiotic observed in vivo, is likely worth consideration.

As mentioned in our response to comment 9 above, we have simultaneously tracked the abundance of *P. aeruginosa* and *S. maltophilia* strains in our cross-protection experiment for select antibiotic concentrations. To be able to perform this experiment, we had to label two extremely-drug-resistant strains of S. maltophilia with an antibiotic resistance marker that allowed us to quantify them in mixtures with *P. aeruginosa*. Our results can be found in Fig. S3 of our revised manuscript and, in a nutshell, show that ceftazidime treatment leads to eradication of both *P. aeruginosa* and *S. maltophilia* when disulfide bond formation is impaired in *S. maltophilia*.

The following text was added to address the questions of the reviewer:

“Due to the naturally different growth rates of these two species (*S. maltophilia* grows much slower than *P. aeruginosa*) especially in laboratory conditions, the protocol we followed [1] requires *S. maltophilia* to be grown for 6 hours prior to co-culturing it with *P. aeruginosa*. To ensure that at this point in the experiment our two *S. maltophilia* strains, with and without dsbA, had grown comparatively to each other, we determined their cell densities (Fig. S3A). We found that S. maltophilia AMM dsbA dsbL had grown at a similar level as the wild-type strain, and both were at a higher cell density [~10^7^ colony forming units (CFUs)] compared to the *P. aeruginosa* PA14 inoculum (5 x 10^4^ CFUs)” (lines 353-361 of the revised manuscript).

“To ensure that ceftazidime treatment leads to eradication of both *P. aeruginosa* and *S. maltophilia* when disulfide bond formation is impaired in *S. maltophilia*, we monitored the abundance of both strains in each synthetic community for select antibiotic concentrations (Fig. S3B). In this experiment we largely observed the same trends as in Fig. 4E. At low antibiotic concentrations, for example 4 μg/mL of ceftazidime, *S. maltophilia* AMM is fully resistant and thrives, thus outcompeting *P. aeruginosa* PA14 (dark pink and dark blue bars in Fig. S3B). The same can also be seen in Fig. 4E, whereby decreased *P. aeruginosa* PA14 CFUs are recorded. By contrast *S. maltophilia* AMM dsbA dsbL already displays decreased growth at 4 μg/mL of ceftazidime because of its non-functional L1-1 enzyme, allowing comparatively higher growth of *P. aeruginosa* (light pink and light blue bars in Fig. S3B). Despite the competition between the two strains, *P. aeruginosa* PA14 benefits from *S. maltophilia* AMM’s high hydrolytic activity against ceftazidime, which allows it to survive and grow in high antibiotic concentrations even though it is not resistant (see 128 μg/mL; dark pink and dark blue bars in Fig. S3B). In stark opposition, without its disulfide bond in *S. maltophilia* AMM dsbA dsbL, L1-1 cannot confer resistance to ceftazidime, resulting in killing of *S. maltophilia* AMM dsbA dsbL and, consequently, also of *P. aeruginosa* PA14 (see 128 μg/mL; light pink and light blue bars in Fig. S3B).

The data presented here show that, at least under laboratory conditions, targeting protein homeostasis pathways in specific recalcitrant pathogens has the potential to not only alter their own antibiotic resistance profiles (Fig. 3 and 4A-D), but also to influence the antibiotic susceptibility profiles of other bacteria that co-occur in the same conditions (Fig. 5). Admittedly, the conditions in a living host are too complex to draw direct conclusions from this experiment. That said, our results show promise for infections, where pathogen interactions affect treatment outcomes, and whereby their inhibition might facilitate treatment” (lines 381406 of the revised manuscript).

(11) Regarding the role of microbial interactions in CF and other disease/infection contexts, the authors should temper their descriptions in accordance with citations provided. As an example, lines 96-99: "For example, in the CF lung, highly drug-resistant *S. maltophilia* strains actively protect susceptible *P. aeruginosa* from β-lactam antibiotics [12], and ultimately facilitate the evolution of β-lactam resistance in *P. aeruginosa* [14]."Neither citation provided here attests to Sm protection of Pa "in the CF lung". Both papers use a simplified in vitro co-culture model to assess Sm protection of Pa from antibiotics and the evolution of Pa antibiotic resistance in the presence or absence of Sm, respectively. In the latter case, it should also be noted that while the authors observed somewhat faster Pa resistance evolution in one co-culture condition, they did not observe it in the other, and that resistance evolution in general was observed regardless of co-culture condition. There are also statements in the ultimate and penultimate paragraphs of the Discussion section that repeat these points. The authors could re-frame this aspect of their investigation as part of a working hypothesis related to potential interactions of these pathogens, and should appropriately caveat what is and is not known from in vitro and in vivo/clinical work.

Thank you for your comment. You are entirely correct. We have amended the test throughout our revised manuscript to avoid overstating these finding and to be clear about the fact that they originate from experimental studies. Please find below representative examples of such passages:

“In particular, some antibiotic resistance proteins, like β-lactamases, which decrease the quantities of active drug present, function akin to common goods, since their benefits are not limited to the pathogen that produces them but can be shared with the rest of the bacterial community. This means that their activity enables pathogen cross-resistance when multiple species are present [1,31], something that was demonstrated in recent work investigating the interactions between pathogens that naturally co-exist in CF infections. More specifically, it was shown that in laboratory co-culture conditions, highly drug-resistant *S. maltophilia* strains actively protect susceptible *P. aeruginosa* from β-lactam antibiotics [1]. Moreover, this crossprotection was found to facilitate, at least under specific conditions, the evolution of β-lactam resistance in *P. aeruginosa* [32]” (lines 47-57 of the revised manuscript).

“The antibiotic resistance mechanisms of *S. maltophilia* impact the antibiotic tolerance profiles of other organisms that are found in the same infection environment. S. maltophilia hydrolyses all β-lactam drugs through the action of its L1 and L2 β-lactamases [7,8]. In doing so, it has been experimentally shown to protect other pathogens that are, in principle, susceptible to treatment, such as *P. aeruginosa* [1]. This protection, in turn, allows active growth of otherwise treatable *P. aeruginosa* in the presence of complex β-lactams, like imipenem [1], and, at least in some conditions, increases the rate of resistance evolution of *P. aeruginosa* against these antibiotics [32]” (lines 332-340 of the revised manuscript).

(12) Regarding the role of *S. maltophilia* in CF disease, the authors should either discuss clinical associations more completely or note the conflicting data on its role in disease. As an example, lines 84-87: "As a result, the standard treatment option, i.e., broad-spectrum βlactam antibiotic therapy, constitutes a severe risk for CF patients carrying both *P. aeruginosa* and *S. maltophilia* [10,11], creating an urgent need for antimicrobial approaches that will be effective in eliminating both pathogens."It is unclear how this treatment results in a "severe risk" for CF patients colonized by both Sm and Pa. Citation 10 suggests an association between anti-pseudomonal antibiotic use and increased prevalence of Sm, but neither citation supports a worsening clinical outcome from this treatment. Citation 10 further notes that clinical scores between Sm-positive and control cohorts could not be distinguished statistically. Citation 11 is a review that makes note of this conflicting data regarding Sm, including reference to a more recent (at the time) result using multivariate analysis showing no independent affect of Sm on survival.The above point similarly applies to other statements in the manuscript, for example at lines 266-267: "Considering the contribution of *S. maltophilia* strains to treatment failure in CF lung infections [8,10,11][...]" As well as lines 79-80: "Pulmonary exacerbations and severe disease states are also associated with the presence of S. maltophilia [8]"Again, the provided citations do not support the implication that Sm specifically 'contributes to treatment failure in CF lung infections' or that Sm is specifically associated with severe disease states. In addition to the previously discussed citations, citation 8 describes broad "pulmotypes" composed of 10 species/genera that could be associated with particular clinical (e.g., exacerbation) or treatment (e.g., antibiotic therapy) characteristics, but these cannot, without further analysis, be associated with, or causally linked to, a specific pathogen. While pulmotype 2 in citation 8 was associated with a more severe clinical state and appeared to have the highest relative abundance of Sm compared to other pulmotypes, Sm was not identified (Figure 4A) as an independent factor that distinguishes between moderate and severe disease, unlike Pa and some anaerobes (4F-H). The authors also observed that decreasing relative abundance of Pa, in particuar, is correlated with subsequent exacerbation, but did not correlate this with the presence of any other species or genera. Again, this should be re-framed with the appropriate caveat that this is a hypothesis with possible clinical significance.Several suggested papers are included below on Sm association with clinical characteristics to incorporate into the manuscript if the authors choose to do so:
https://doi.org/10.1177/14782715221088909

https://doi.org/10.1016/j.prrv.2010.07.003

https://doi.org/10.1016/j.jcf.2013.05.009
https://doi.org/10.1002/ppul.23943

https://doi.org/10.1002/14651858.CD005405.pub2

https://doi.org/10.1164/rccm.2109078
http://dx.doi.org/10.1136/thx.2003.017707

https://erj.ersjournals.com/content/23/1/98.short

Thank you for your comment. You are entirely correct. We have amended the test throughout our revised manuscript to avoid overstating the role of *S. maltophilia* in CF infections and to reference additional relevant works in the literature. Please find below representative examples of such passages:

“On the other hand, CF microbiomes are increasingly found to encompass *S. maltophilia* [2-4], a globally distributed opportunistic pathogen that causes serious nosocomial respiratory and bloodstream infections [5-7]. S. maltophilia is one of the most prevalent emerging pathogens [6] and it is intrinsically resistant to almost all antibiotics, including β-lactams like penicillins, cephalosporins and carbapenems, as well as macrolides, fluoroquinolones, aminoglycosides, chloramphenicol, tetracyclines and colistin. As a result, the standard treatment option for lung infections, i.e., broad-spectrum β-lactam antibiotic therapy, is rarely successful in countering S. maltophilia [7,8], creating a definitive need for approaches that will be effective in eliminating both pathogens” (lines 33-41 of the revised manuscript).

“Of the organisms studied in this work, *S. maltophilia* deserves further discussion because of its unique intrinsic resistance profile. The prognosis of CF patients with S. maltophilia lung carriage is still debated [4,9-16], largely because studies with extensive and well-controlled patient cohorts are lacking. This notwithstanding, the therapeutic options against this pathogen are currently limited to one non-β-lactam antibiotic-adjuvant combination, , which is not always effective, trimethoprim-sulfamethoxazole [17-20], and a few last-line β-lactam drugs, like the fifth-generation cephalosporin cefiderocol and the combination aztreonam-avibactam. Resistance to commonly used antibiotics causes many problems during treatment and, as a result, infections that harbor S. maltophilia have high case fatality rates [7]. This is not limited to CF patients, as S. maltophilia is a major cause of death in children with bacteremia [5]” (lines 440-450 of the revised manuscript).

**Reviewer #3 (Recommendation For the Authors):**
(1) The referencing of supplemental figures does not follow a sequential order. For example, Figure S2 appears in the text before S1. The sequential ordering of figure numbers improves the readability and can be considered while editing the manuscript for revision.

Thank you for this comment. This is amended in our revised manuscript and supplemental figures and files are cited in order.

(2)It will be useful to provide a brief description of ambler classes since these are important to study design (for a broader audience).

Thank you for this suggestion. This has been added and can be found in lines 91-101 of the revised manuscript.

(3) The rationale for using K12 strain for *E. coli* should be provided. It appears that is a model system that is well established in their lab, but a scientific rationale can be listed. Maybe this strain does not have any lactamases in its genome other than the one being expressed as compared to pathogenic *E. coli*?

Thank you for this suggestion. This has been added and can be found in lines 104-106 of the revised manuscript.

(4) The reviewers used worm model to test their observations, which is relevant. Given the significant implications of their work in overcoming resistance to clinically used antibiotics and availability of already generated dsbA mutants in clinical strains, it will be useful to investigate survival in animal models or at least wound models of Pseudomonas infections. The reviewer does not deem this necessary, but it will significantly increase the impact of their seminal work.

Thank you for this comment. We appreciate the sentiment, and we would have liked to be able to perform experiments in a murine model of infection. There are several reasons that made this not possible, and as a result we used G. mellonella as an informative preliminary in vivo infection model. The DSB proteins have been shown to play a central role in bacterial virulence. Because of this our *P. aeruginosa* and *S. maltophilia* mutant strains are not efficient in establishing an infection, even in a wound model. This could be overcome had we been able to use the chemical inhibitor of the DSB system in vivo, however this also is not possible This is due to the fact that the chemical compound that we use to inhibit the function of DsbA acts on DsbB. Inhibition of DsbB blocks the re-oxidation of DsbA and leads to its accumulation in its inactive reduced form. However, the action of the inhibitor can be bypassed through reoxidation and re-activation of DsbA by small-molecule oxidants such as L-cystine, which are abundant in rich growth media or animal tissues. This makes the inhibitor only suitable for in vitro assays that can be performed in minimal media, where the presence of small-molecule oxidants can be strictly avoided, but entirely unsuitable for an insect or a vertebrate animal model.